# Multilevel multifidelity Monte Carlo methods for assessing uncertainty in coastal flooding

Mariana C. A. Clare[1, 3], Tim W. B. Leijnse[2], Robert T. McCall[2], Ferdinand L. M. Diermanse[2], Colin J. Cotter[1], and Matthew D. Piggott[1]

[1]Imperial College London, UK
[2]Deltares, NL
[3]ECMWF, Bonn, DE

**Correspondence:** Mariana Clare (m.clare17@imperial.ac.uk)

**Abstract.** When choosing an appropriate hydrodynamic model, there is always a compromise between accuracy and computational cost, with high fidelity models being more expensive than low fidelity ones. However, when assessing uncertainty, we can use a multifidelity approach to take advantage of the accuracy of high fidelity models and the computational efficiency of low fidelity models. Here, we apply the *multilevel multifidelity* Monte Carlo method (MLMF) to quantify uncertainty by computing statistical estimators of key output variables with respect to uncertain input data, using the high fidelity hydrodynamic model XBeach and the lower fidelity coastal flooding model SFINCS (Super-Fast INundation of CoastS). The multilevel aspect opens up the further advantageous possibility of applying each of these models at multiple resolutions. This work represents the first application of MLMF in the coastal zone and one of its first applications in any field. For both idealised and real-world test cases, MLMF can significantly reduce computational cost for the same accuracy compared to both the standard Monte Carlo method and to a multilevel approach utilising only a single model (the multilevel Monte Carlo method). In particular, here we demonstrate using the case of Myrtle Beach, USA, that this improvement in computational efficiency allows in-depth uncertainty analysis to be conducted in the case of real-world coastal environments – a task that would previously have been practically unfeasible. Moreover, for the first time, we show how an inverse transform sampling technique can be used to accurately estimate the cumulative distribution function (CDF) of variables from the MLMF outputs. MLMF based estimates of the expectations and the CDFs of the variables of interest are of significant value to decision makers when assessing uncertainty in predictions.

## 1 Introduction

Throughout history, coastal zones have been attractive regions for human settlement and leisure due to their abundant resources and the possibilities they offer for commerce and transport. Nevertheless, living in coastal zones has always come with the risk of coastal flooding hazards, for example, from storm surges as well as wave run-up and overtopping. Hydrodynamic models can simulate these hazards but these predictions are often uncertain (Athanasiou et al., 2020), due to uncertainties in input data as well as in the hydrodynamic models themselves. Standard practice to assess these uncertainties is to express these uncertain inputs/parameters using probability distributions. The uncertainty can then be assessed by sampling from these distributions

and computing key output diagnostics such as the mean and variance of key variables of interest and/or the probability of a hazard event occurring (see for example Kalyanapu et al. (2012) and Wang et al. (2017) where they are used to assess impacts from flooding and hurricanes respectively). These statistics can all be expressed as expectations and can be estimated by computing statistical estimators. The most straightforward approach to compute such an estimator is to apply the standard form of the Monte Carlo method. For a given model $X$ and an uncertain input parameter $\alpha$, the Monte Carlo estimator $\hat{f}(X)$ for the expectation $\mathbb{E}[f(X(\alpha))]$ is given by

$$\hat{f}(X) = \frac{1}{N} \sum_{n=1}^{N} f(X(\alpha^{(n)})) \tag{1}$$

where $\alpha^{(n)}$ are $N$ independent samples taken from the distribution of the uncertain input parameter. However, this method requires $O(\epsilon^{-2})$ model runs to achieve an accuracy $\epsilon$ (Caflisch, 1998), which can easily make it prohibitively computationally expensive, especially given the high computational cost of accurate coastal models. In existing research such as Callaghan et al. (2013), low fidelity models are used to solve the issue of high computational cost with Monte Carlo methods, but this leads to less accurate results.

We take an alternative approach and instead compute statistical estimators using the relatively novel *multilevel multifidelity* Monte Carlo (MLMF) method, developed in Geraci et al. (2015), which combines results from a high fidelity and a low fidelity model. MLMF takes advantage of the accuracy of high fidelity models and the computational efficiency of lower fidelity ones to produce accurate yet computationally feasible uncertainty analyses. It further improves computational efficiency by using the hierarchy of model resolutions approach, similar to that used in the multilevel Monte Carlo method (MLMC) (Giles, 2008). Research into MLMF is still in its infancy, and this work represents the first application of MLMF in the coastal zone. It has, however, already been successfully applied in aerospace research (Geraci et al., 2017) and cardiology (Fleeter et al., 2020). Note that MLMC is also a fairly novel method, but it has already been successfully applied to coastal zones in Clare et al. (2021), a promising indication that MLMF will be similarly successful in this field.

MLMF does not aim to improve the accuracy relative to using a standard Monte Carlo method on the high fidelity model, but to instead use a lower fidelity model to accelerate the approach and thus make uncertainty studies computationally feasible. Therefore the key to the successful application of MLMF is choosing an accurate high fidelity model and an appropriate lower fidelity model, which reasonably approximates the high fidelity one. Coastal flood modelling is therefore an ideal field on which to apply MLMF because there exist a large number of high fidelity but computationally expensive full physics models such as XBeach (Roelvink et al., 2009), SWASH (Simulating WAves till SHore) (Zijlema et al., 2011), or MIKE21 (Warren and Bach, 1992), and lower fidelity computationally cheaper reduced physics models such as SFINCS (Super-Fast INundation of CoastS) (Leijnse et al., 2021), LISFLOOD-FP (Bates et al., 2010) or SBEACH (Storm-Induced BEAch CHange) (Larson and Kraus, 1989). Furthermore, this work provides an interesting example of a framework for combining lower and high fidelity models in an area where there is already a lot of research into combining different fidelity models (for example Callaghan et al., 2013; Leijnse et al., 2021).

In our work, we choose the depth-averaged finite-volume based coastal ocean model XBeach as our high fidelity model because it can parameterise unresolved wave propagation, such as wind-driven wave fields, and has been successfully used

numerous times in the coastal zone to simulate wave propagation and flow including, for example, in Roelvink et al. (2018) and de Beer et al. (2020). For our lower fidelity model, we use the hydrodynamic model SFINCS because of its ability to simulate the relevant processes for compound coastal flooding (Leijnse et al., 2021). Note that to maximise computational efficiency, SFINCS does not explicitly solve for short wavelength wind-driven waves internally but instead these can be provided in the form of a prescribed forcing. The computational efficiency of SFINCS has already been favourably compared to XBeach in numerous test cases (see Leijnse, 2018; Leijnse et al., 2021), where SFINCS is shown to be significantly cheaper than XBeach, with acceptable differences in accuracy. Despite this model choice, we emphasise that we implement our MLMF algorithm using a model-independent Python wrapper developed in our work, which could easily be applied to other coastal ocean models in future research. Note further that whilst investigating the accuracy of the specific models used is beyond the scope of this work, this wrapper approach means that the numerous verification and validation studies conducted with XBeach and SFINCS still hold for our work (for example McCall et al., 2010; Riesenkamp, 2011; Roelvink et al., 2018; Leijnse et al., 2021).

The aim of this work is to explore how MLMF can be applied to complex hydrodynamic coastal ocean models to investigate within a reasonable timeframe, the impact of a variety of uncertain input parameters, such as wave height and bed slope angle, whilst maintaining accuracy relative to the standard Monte Carlo method. We apply MLMF to both idealised and real-world test cases, some of which would have been impractical and unrealistic to run using standard Monte Carlo methods due to huge computational costs. In many of these test cases, we conduct a valuable spatial uncertainty analysis of the coastal flooding, by calculating the expected value of output variables simultaneously at multiple locations. Like other Monte Carlo type methods, MLMF quantifies uncertainty by computing estimators of the expected value of key output variables with respect to uncertain input parameters. However, in this work we also modify the inverse transform sampling method from Gregory and Cotter (2017) to develop a novel method to generate Cumulative Distribution Functions (CDFs) from MLMF outputs. This provides information allowing practitioners to determine the probability of a variable exceeding a certain value, which can be of more interest than the expected value.

The remainder of this work is structured as follows: in Section 2 we outline the methodology for applying MLMF to the coastal flood models and the relevant MLMF theory; in Section 3, we apply MLMF with SFINCS and XBeach to idealised and real-world test cases to estimate both the expected value and the cumulative distribution function for the considered output variables; in Section 4, we discuss extensions to the MLMF methodology; and, finally, in Section 5, we conclude this work.

## 2 Methodology: Applying the multilevel multifidelity Monte Carlo method (MLMF) to assess uncertainty in coastal flooding

As discussed in Section 1, Monte-Carlo type methods can be used to assess uncertainty by estimating the expectations of functions of an input random variable. In our model scenario, the input random variable is some source of uncertainty, such as the friction coefficient, and the function involves running our numerical model and computing values such as the water elevation height at specific locations, from the model output. These estimates could be calculated using the standard Monte Carlo approach, but this is computationally expensive due to the need to run large numbers of model simulations to obtain an

appropriate accuracy (see Eq. 1 and the discussion below it). The computational cost of running the model can be reduced by either coarsening the grid resolution or using a less complex model, or, in the case of this work, making use of both approaches by using the multilevel multifidelity Monte Carlo method (MLMF).

Using a coarse grid and/or simpler model gives an estimate which is cheap to compute but (more) incorrect and thus has an
error. This error can be corrected by estimating the difference between the low and high fidelity models and/or the different resolutions, and adding these on to the cheaply computed expectation. Key to the approach is the observation that estimating the difference requires fewer simulations than computing the full estimate, because the variance of the correction is (hopefully) smaller than the variance of the outputs. For the different grid resolutions, the correction is done by the telescoping sum of the multilevel Monte Carlo method (MLMC), while for the different fidelity models, the correction is done by control variate
formulae. The challenge is composing these approaches so that we can do both, which is what MLMF seeks to do.

The theory for MLMF is the focus of Section 2.1, whilst details on the control variate multifidelity approaches and MLMC can be found in Appendix A and B respectively. As described in Geraci et al. (2015), the standard MLMF approach cannot estimate the probability of an output variable exceeding a certain value. The latter is often also of significant interest for flooding problems and thus in Section 2.2, we present novel theory to extend MLMF for the estimation of probabilities. The
implementation and application of the MLMF method in this work is then described in Section 2.3 and we conclude this methodology section with a brief remark on different methods to assess uncertainty in Section 4.

## 2.1 Multilevel multifidelity Monte Carlo method (MLMF)

MLMF seeks to improve the efficiency of uncertainty analyses by running fewer simulations at the more expensive finer resolutions than at the cheaper coarser resolutions and by running fewer high fidelity model simulations than low fidelity
ones (see Figure 1). In this section, we describe the theory for the standard MLMF approach, following Geraci et al. (2015) throughout. A pictorial representation of this algorithm is shown in Figure 2 and a full statement of the algorithm is included at the end of the section.

To fix ideas, we consider a hypothetical scenario, where the variable of interest is the water elevation height at a given location after a given time and the uncertain parameter is the friction coefficient which we assume follows a normal distribution. The
desired grid resolution in our model is $\Delta x = 5000/2^{10} \ (\approx 5) \, \text{m}$. Note that this hypothetical scenario is similar to the example used as the first test case in this work. Moreover, throughout this work, we use HF and LF to denote the high fidelity XBeach model and low fidelity SFINCS model respectively.

We denote the MLMF estimator for the water depth at the finest grid resolution $L$ as $\hat{Q}_{M_L}^{HF,CV}$. Here the finest grid resolution is the grid resolution we would like to evaluate our model at; for our hypothetical scenario the finest grid resolution is
$\Delta x = M/2^L = 5000/2^{10} \ (\approx 5) \, \text{m}$. Note that following standard notation, $\hat{\cdot}$ denotes that $\hat{Q}_{M_L}^{HF,CV}$ is an estimator. An estimator represents the rule for calculating an estimate of a variable of interest given data. In our hypothetical scenario, the estimator is the rule, the variable of interest is the water elevation height, the data is our model runs and the estimate is then the numerical approximation of the mean water elevation that we obtain using our model runs. For MLMF, the rule for the estimator is a combination of the multilevel MLMC estimator (B3) with the multifidelity control variate (A1). The multilevel part of the

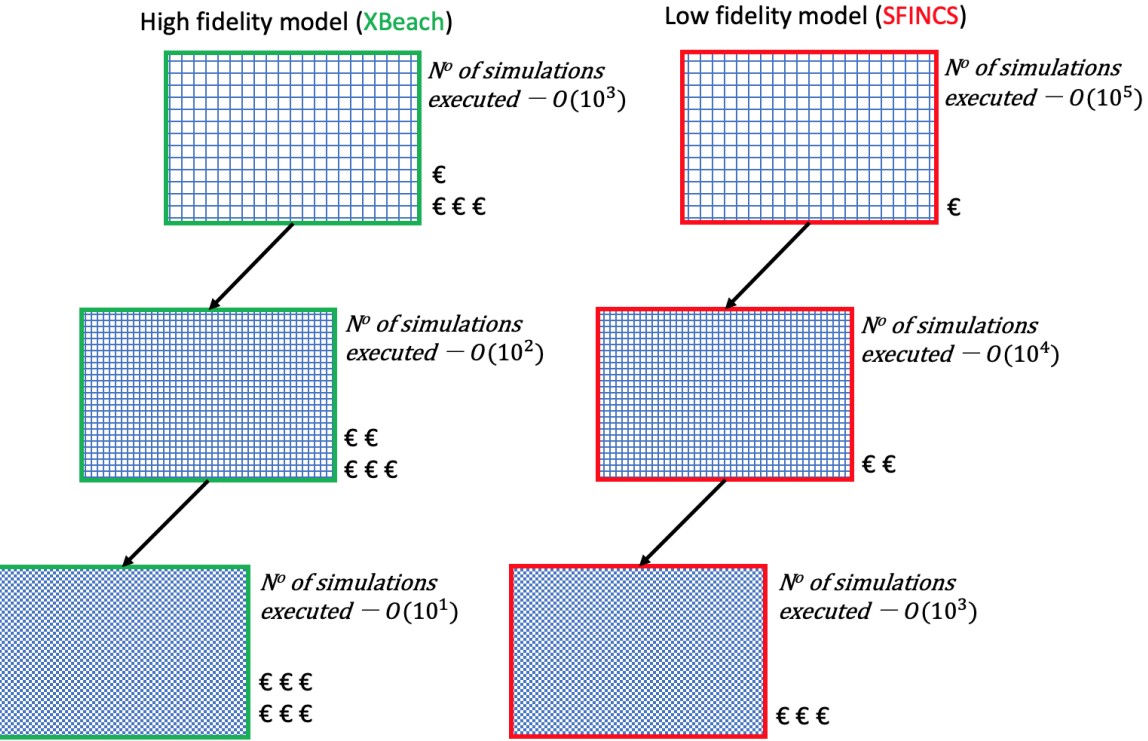

**Figure 1.** Example illustration of how MLMF's multifidelity multilevel approach using SFINCS and XBeach models on different grid resolutions results in computational cost savings. Note the € symbol indicates the order of magnitude of the computational cost for a single simulation with this model at this grid resolution *i.e.* €€ indicates $O(10^2)$ seconds for a single simulation. The orders of time and number of scenarios are approximately those for the Myrtle Beach test case in Section 3.3.

estimator uses linearity of expectations (see (B1)) to construct the following telescoping sum

$$\hat{Q}_{M_L}^{HF,CV} = \hat{Q}_{M_{l_\mu}}^{HF,CV} + \sum_{l=l_\mu+1}^{L} \left[ \hat{Q}_{M_l}^{HF,CV} - \hat{Q}_{M_{l-1}}^{HF,CV} \right], \tag{2}$$

where $M_l$ denotes different resolutions at which the estimator is evaluated, with $l_\mu$ being the coarsest resolution. In our hypothetical scenario, the estimator is evaluated at resolutions of $[5000/2^4, 5000/2^5, 5000/2^6, 5000/2^7, 5000/2^8, 5000/2^9, 5000/2^{10}]$m. Eq. (2) finds the multilevel multifidelity estimate of water elevation at the finest resolution by calculating the multifidelity estimate at the coarsest resolution $(5000/2^4)$, adding to this the difference between the multifidelity estimates at the coarsest

resolution $(5000/2^4)$ and the slightly finer resolution $(5000/2^5)$ etc., up to and including the second finest and finest resolutions pair of $5000/2^9$ and $5000/2^{10}$. By the linearity of expectations, the sum of these differences is an estimate for the expected value of the water elevation on the finest resolution that is as accurate as simply calculating a standard Monte Carlo estimate on the finest resolution. The advantage is that calculating the estimate using this approach is less computationally

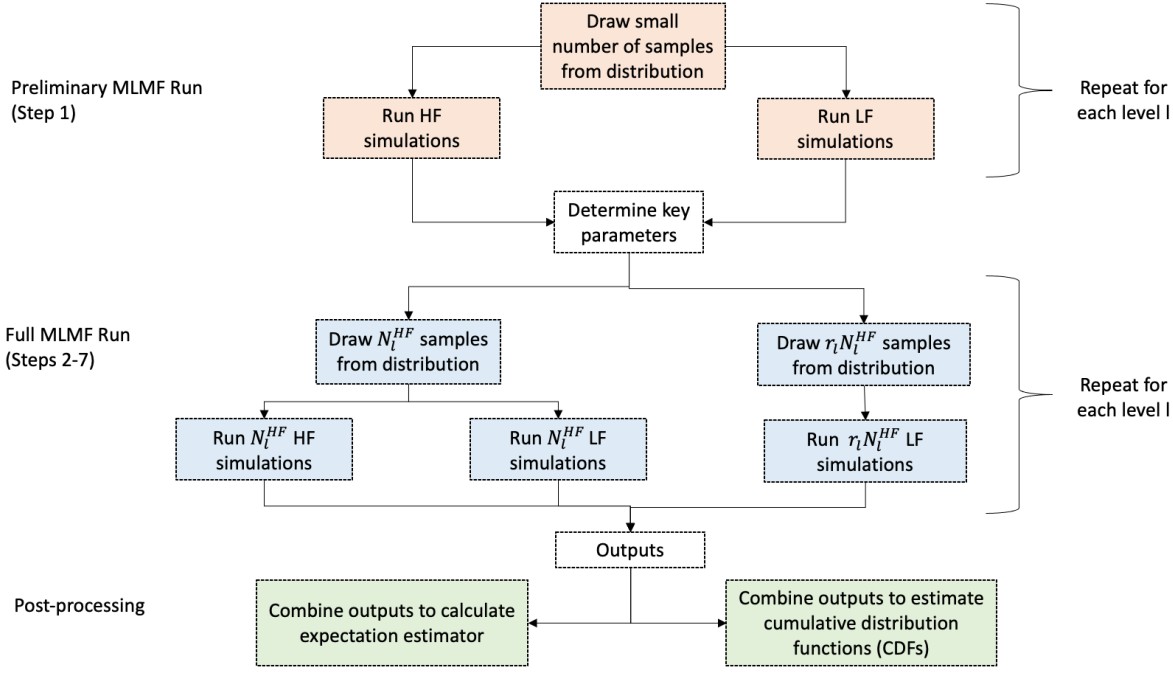

**Figure 2.** Flow chart of multi-model approach to MLMF using HF (XBeach) and LF (SFINCS).

expensive than using the standard Monte Carlo approach because the width of the distribution of the model outputs at each resolution $X_l$ is much larger than the width of the distribution of the *difference* between the outputs ($X_l - X_{l-1}$). Figure 3 illustrates this for two resolutions of the hypothetical scenario computed using XBeach. The narrower the distribution (*i.e.* the smaller the variance) the fewer samples are needed to estimate its mean (see Figure 9 for example). Note that the distribution of the difference is very narrow in this example; for more complex cases it may be wider, although it should still remain narrower than the distribution of the individual outputs.

Combining multilevel estimators with the multifidelity control variate (A1), the full rule for the MLMF estimator is

$$\hat{Q}_{M_L}^{HF,CV} = \sum_{l=l_\mu}^{L} \left( \hat{Y}_{M_l}^{HF} + \alpha_l \left( \hat{Y}_{M_l}^{LF} - \hat{E}\left[ Y_{M_l}^{LF} \right] \right) \right), \tag{3}$$

where

$$\hat{Y}_{M_l}^* = \begin{cases} N_{l_\mu}^{-1} \sum_{i=1}^{N_{l_\mu}} X_{l_\mu}^{(i)} & l = l_\mu, \\ N_l^{-1} \sum_{i=1}^{N_l} \left( X_l^{(i)} - X_{l-1}^{(i)} \right) & l > l_\mu, \end{cases} \tag{4}$$

where the superscript $*$ here can indicate results from either XBeach (HF) or SFINCS (LF). In our hypothetical scenario, $X_l$ is the water elevation height from the model run using a grid resolution of $\Delta x = 5000/2^l$ m, with $X_{l-1}$ being the same but for a grid resolution of $\Delta x = 5000/2^{l-1}$ m. For each difference pair, $(i)$ denotes that the value sampled from the normal distribution

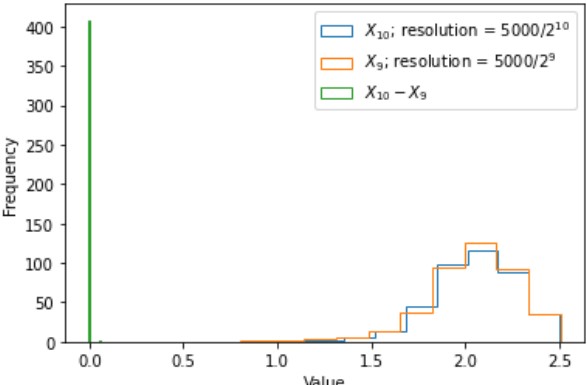

**Figure 3.** Distribution of outputs generated by XBeach for the hypothetical scenario at the finest resolution and at the second finest resolution considered, as well as the distribution of the difference between these output values. Note that the distribution for the difference between the output values is much narrower, meaning fewer samples are required to get a good estimate of the mean. For reference, in this hypothetical scenario the variance at resolution $\Delta x = 5000/2^9$m is 0.0587; the variance at resolution $\Delta x = 5000/2^{10}$m is 0.0580; and the variance of the difference between outputs at $[5000/2^9, 5000/2^{10}]$ is 9.82e-06.

for the uncertain friction coefficient is the same for both the finer resolution $X_l^i$ simulation and the coarser resolution $X_l^{i-1}$ simulation. Thus, $\hat{Y}_{M_l}^*$ is the mean of the difference between two model runs conducted at different resolutions with the same random number (*i.e.* value sampled from the distribution) used for friction for each pair. Moreover, for each $(i)$, the same random number is used for the XBeach model pair and the SFINCS model pair, *i.e.* the same random numbers are used to construct both $\hat{Y}_{M_l}^{HF}$ and $\hat{Y}_{M_l}^{LF}$. Note constructing estimators like this means that the coarsest level, $l_\mu$ is left without a pair and therefore $\hat{Y}_{M_{l_\mu}}^*$ is just the mean of the model runs conducted at the coarsest resolution. Note further that, although not strictly necessary, here we choose to run both SFINCS and XBeach at the same resolutions, as it seems sensible to assume that this will maximise correlation between the outputs at each level.

The other terms in (3) come from the multifidelity estimator. The notation $\hat{E}[\cdot]$ denotes the estimator for the expected value – statistically speaking we cannot know the actual expected value ($\mathbb{E}$) of $Y_{M_l}^{LF}$ because this would require knowing the exact distribution of $Y_{M_l}^{LF}$. Thus, the best we can do is calculate an estimate of the expected value using data from SFINCS runs at different resolutions, *i.e.* use an estimator. This subtlety is discussed in more detail in Appendix A. Finally $\alpha_l$ is a coefficient which weights the SFINCS model outputs and is defined as

$$\alpha_l = -\rho_l \sqrt{\frac{\text{Var}(\hat{Y}_{M_l}^{HF})}{\text{Var}(\hat{Y}_{M_l}^{LF})}}, \tag{5}$$

where $\rho_l$ is the Pearson's correlation coefficient. In our hypothetical scenario, $\rho_l$ is the correlation between the water elevation calculated by XBeach and that calculated by SFINCS at each resolution $l$. We refer the reader to Appendix A for more details on multifidelity estimators.

To make calculating the variance of the water depth estimator simpler, we follow standard practice throughout and independently sample the values for the friction coefficient for each $\hat{Y}_{M_l}$. Hence the variance of the MLMF estimator is

$$\mathrm{Var}\left[\hat{Q}_{M_L}^{HF,CV}\right] = \sum_{l=l_\mu}^{L} \left(N_l^{HF}\right)^{-1} \mathrm{Var}\left[\hat{Y}_l^{HF}\right]\left(1 - \frac{r_l}{1+r_l}\rho_l^2\right), \tag{6}$$

using independence. Here $N_l^{HF}$ is the number of XBeach (HF) simulations required at level $l$ to compute $\hat{Y}_{M_l}^{HF}$ (which is also the number of SFINCS (LF) simulations required to compute $\hat{Y}_{M_l}^{LF}$), and $r_l$ is the factor of extra SFINCS simulations required to compute $\hat{E}\left[Y_{M_l}^{LF}\right]$. Note that, throughout this work and for simplicity, we refer to $N_l^{HF}$ as the number of XBeach simulations required, because the total number of SFINCS simulations required is the combined quantity $(1+r_l)N_l^{HF}$, and not just $N_l^{HF}$.

Because $\rho_l^2 < 1$ by definition of a correlation coefficient, equation (6) shows that the greater the correlation between the two models, the greater the reduction in the variance of the estimator. We thus seek to maximise this correlation. Geraci et al. (2017) show that, because the multifidelity control variate is unbiased, correlation can be artificially increased by modifying the estimator $\hat{Y}_l^{LF}$ using

$$\mathring{Y}_l^{LF} = \gamma_l \hat{X}_l^{LF} - \hat{X}_{l-1}^{LF}, \tag{7}$$

where the modification factor $\gamma_l$ adds an extra degree of freedom to maximise the correlation. Therefore, instead of (3), we use

$$\hat{Q}_{M_L}^{HF,CV} = \sum_{l=l_\mu}^{L} \left(\hat{Y}_{M_l}^{HF} + \alpha_l\left(\mathring{Y}_l^{LF} - \hat{E}\left[\mathring{Y}_l^{LF}\right]\right)\right), \tag{8}$$

and the new correlation coefficient $\mathring{\rho}_l^2$ is dependent on $\gamma_l$ and is equal to

$$\mathring{\rho}_l^2 = \rho_l^2 \frac{\mathrm{Cov}^2\left(\hat{Y}_l^{HF},\mathring{Y}_l^{LF}\right)\mathrm{Var}\left[\hat{Y}_l^{LF}\right]}{\mathrm{Cov}^2\left(\hat{Y}_l^{HF},\hat{Y}_l^{LF}\right)\mathrm{Var}\left[\mathring{Y}_l^{LF}\right]}, \tag{9}$$

where we correct a typographical error in the formula given in Geraci et al. (2017). By differentiating (9) with respect to $\gamma_l$, we find the correlation is maximised when

$$\gamma_l = \frac{\mathrm{Cov}\left(\hat{Y}_l^{HF},X_{l-1}^{LF}\right)\mathrm{Cov}\left(X_l^{LF},X_{l-1}^{LF}\right) - \mathrm{Var}\left[X_{l-1}^{LF}\right]\mathrm{Cov}\left(\hat{Y}_l^{HF},X_l^{LF}\right)}{\mathrm{Var}\left[X_l^{LF}\right]\mathrm{Cov}\left(\hat{Y}_l^{HF},X_{l-1}^{LF}\right) - \mathrm{Cov}\left(\hat{Y}_l^{HF},X_l^{LF}\right)\mathrm{Cov}\left(X_l^{LF},X_{l-1}^{LF}\right)}. \tag{10}$$

Note that when using the modified estimator (7) the formulae previously stated in this section remain the same but $\hat{Y}_l^{LF}$ and $\rho_l$ are replaced with $\mathring{Y}_l^{LF}$ and $\mathring{\rho}_l$, respectively, in all formulae.

Finally, using (A3) and (B6), the overall cost of the MLMF algorithm (*i.e.* finding the water elevation at grid resolution $L$) is

$$C = \sum_{l=l_\mu}^{L} N_l^{HF}\left(C_l^{HF} + C_l^{LF}(1+r_l)\right). \tag{11}$$

In order to obtain the optimum values for $N_l^{HF}$ and $r_l$ in (6), we minimise this cost with respect to the variance constraint

$$\text{Var}\left[\hat{Q}_{M_L}^{HF,CV}\right] < \epsilon^2/2. \tag{12}$$

which results in the following optimum formula for the factor of extra SFINCS simulations

$$r_l = -1 + \sqrt{\frac{\mathring{\rho}_l^2}{1 - \mathring{\rho}_l^2}\omega_l}, \tag{13}$$

where $\omega_l = C_l^{HF}/C_l^{LF}$ is the ratio of the cost of running XBeach and SFINCS, and the following optimum formula for the number of XBeach simulations

$$N_l^{HF} = \frac{2}{\epsilon^2}\left[\sum_{k=l_\mu}^{L}\left(\frac{\text{Var}\left[\mathring{Y}_k^{HF}\right]C_k^{HF}}{1 - \mathring{\rho}_l^2}\right)^{1/2}\Lambda_k(r_k)\right]\sqrt{(1 - \mathring{\rho}_l^2)\frac{\text{Var}\left[\mathring{Y}_l^{HF}\right]}{C_l^{HF}}}, \tag{14}$$

where

$$\Lambda_k(r_k) = 1 - \frac{r_k}{1 + r_k}\mathring{\rho}_k^2, \tag{15}$$

and as in (B7), $\epsilon$ should be viewed as a user-defined accuracy tolerance.

Calculating (14) requires estimates of the variance and cost. Therefore we run 50 initial simulations for each model at each resolution (see Step 1 of Algorithm 1) and use the kurtosis to check whether this provides a good enough estimate of the variance. Following Giles (2008), if the kurtosis is less than 100, then we consider our estimate of the variance to be good enough. Note further that if we are interested in the value of the variable of interest at multiple locations, $N_l^{HF}$ must be calculated separately for each location. In the algorithm, we run $\max N_l$ over all locations and then when calculating the estimator (3) at each location, subsample the optimum number for that specific location from the full output.

### 2.1.1 MLMF algorithm

Given the theory outlined above, the MLMF algorithm used in this study is summarised in Algorithm 1.

---

**Algorithm 1** Multilevel Multifidelity Monte Carlo method.

---

1: Estimate the variance and cost of the MLMF estimator, as well as the correlation and cost ratio between the HF and LF models at user-specified levels using an initial estimate for the number of simulations. The same set of random numbers must be used for the HF and LF models

2: Start with $L = l_\mu$

3: Define optimal $N_l^{HF}$ using (14) and $r_l$ using (13) with increased correlation factor (9) when required

4: If the optimal $N_l^{HF}$ is greater than the number of simulations of the HF and LF models from Step 1, evaluate the extra simulations required

5: If the optimal $r_l N_l^{HF}$ is greater than the number of simulations of the LF model after Step 4, evaluate the extra simulations of LF required

6: If the algorithm has not converged and $L < L_{\max}$, set $L$ equal to $L + 1$ and return to Step 3

7: If algorithm converged, or $L \geq L_{\max}$, STOP

---

## 2.2 Cumulative distribution functions

In this section so far, we have described the standard MLMF framework outlined in Geraci et al. (2015), the objective of which is to find the expectation of the output variable of interest. However, the probability of a variable exceeding a certain value is often of significant value in the study of natural hazards. This probability is complicated to estimate because MLMF computes very few values at the finest resolution from which we could build the distribution.

To resolve this, in this work we develop our own novel technique to find the cumulative distribution function (CDF) from the MLMF outputs, using a modified version of the inverse transform sampling method from Gregory and Cotter (2017). The output of a cumulative distribution function, $\mathbb{P}(X \leq x)$, is some value between 0 and 1. Returning to the hypothetical example used throughout this section, $X$ is the water elevation as a variable and $x$ is its value. For evaluating uncertainty, we would like to know the value of $x$ which the water elevation at a given location after a given time is below for $25\%$ of cases, $50\%$ of cases etc. In other words, we are interested in the inverse cumulative distribution function $F^{-1}(u)$, where $u \sim \mathcal{U}[0,1]$ and $F(x) = \mathbb{P}(X \leq x)$. If $F$ is strictly increasing and absolutely continuous, then $x \equiv F^{-1}(u)$ is unique. A simple consistent estimate for $x$ can then be found by sorting the values such that $X^1 < X^2 < ... < X^N$ and then

$$\hat{F}^{-1}(u) = X^{\lceil N \times u \rceil}. \tag{16}$$

In other words, suppose in our hypothetical scenario we have 100 values for the water elevation at a given location after a given time. Then this expression simply says that the value $x$ which the water elevation does not exceed $25\%$ of the time, is the 25th largest value. Gregory and Cotter (2017) show that this estimate is consistent because it converges in probability to $x$ as $N \to \infty$. Note that here converges in probability means that the probability of $X^{\lceil N \times u \rceil}$ being more than a small distance $\epsilon$ from $x$ tends to zero as $N \to \infty$. In Gregory and Cotter (2017), they then use a formula to approximate $F_L^{-1}(u)$ from the MLMC outputs. In this work, we modify that formula to make it applicable for MLMF outputs so that the inverse cumulative

distribution function for MLMF is approximated by

$$
\begin{aligned}
F_L^{-1}(u) \approx\ & R^{HF}(X)_{l_\mu}^{\lceil N_{l_\mu}^{HF} \times u \rceil} + \alpha_{l_\mu} \left( \mathring{R}^{LF}(X)_{l_\mu}^{\lceil N_{l_\mu}^{HF} \times u \rceil} - \hat{E}\left[ \mathring{Y}_l^{LF} \right] \right) \\
& + \sum_{l=l_\mu+1}^{L} \left( R^{HF}(X)_l^{\lceil N_l^{HF} \times u \rceil} - R^{HF}(X)_{l-1}^{\lceil N_{l-1}^{HF} \times u \rceil} \right) \\
& + \sum_{l=l_\mu+1}^{L} \alpha_l \left( \mathring{R}^{LF}(X)_l^{\lceil N_l^{HF} \times u \rceil} - \mathring{R}^{LF}(X)_{l-1}^{\lceil N_l^{HF} \times u \rceil} - \hat{E}\left[ \mathring{Y}_l^{LF} \right] \right),
\end{aligned}
\tag{17}
$$

where $R^{HF}(X)_l^i$ and $\mathring{R}^{LF}(X)_l^i$ represent the $i^{\text{th}}$ order statistic of $X_l$ on each level $l$ of XBeach and modified correlation SFINCS (see 7), respectively. In other words, suppose that in our hypothetical scenario we want to know the value $x$ which the water elevation does not exceed 25% of the time. We then pick the lower quartile value (*i.e.* the value not exceeded 25% of the time at each resolution for both models) and add them together following the rule of the MLMF estimator. Note that, unlike with (B1), there cannot be exact cancellation because using this method means the approximations at each level are no longer unbiased.

## 2.3 Implementation

In this work, we construct our own Python MLMF wrapper around both SFINCS and XBeach to implement the MLMF algorithm. This wrapper can be shared on distributed cores of an HPC cluster to increase efficiency. Given the use of distributed cores, any times quoted in this work are the total simulation times multiplied by the number of cores used. The different steps performed when running the models in the wrapper are illustrated in the flow chart of Figure 2. Note, in particular, that in this wrapper, the models are run and post-processed separately, meaning there is no issue with different input or output formats. Therefore, our MLMF wrapper is model-independent meaning it can be easily applied to other models and applications in further work.

For the models themselves, we use XBeach version 1.23.5526 from the XBeachX release and use the surfbeat mode to simulate the waves approaching the beach (Roelvink et al., 2018). SFINCS is not yet released in the public domain, but we use a version similar to that used in Leijnse et al. (2021).

## 3 Applying MLMF to coastal zone test cases

We can now apply the outlined MLMF algorithm to both idealised and real-world coastal flooding test cases to calculate the expectation of an output variable at multiple locations based on uncertain input data. Note that throughout, for simplicity, we only consider one uncertain input parameter per test case (see Section 4).

In the first 1D test case, the water level is estimated at various locations as a result of a propagating non-breaking wave entering a domain under an uncertain Manning friction coefficient (Section 3.1). In the second 1D test case, the wave run-up height is estimated for a simplified linear beach under the uncertainty of the beach slope (Section 3.2). In the final 2D real world

case of Myrtle Beach, the maximum water depth due to flooding is estimated at various locations influenced by an uncertain offshore water level (Section 3.3).

## 3.1 Non-breaking wave test case

For our first test case, we consider the 1D case of a non-breaking wave propagating over a horizontal plane from Hunter et al. (2005) and Bates et al. (2010), which has already been simulated using SFINCS and XBeach in Leijnse (2018). The domain is initially dry and the wave is generated by imposing a rising water elevation boundary condition and a constant velocity boundary condition ($u(x = 0, t) = 1\mathrm{ms}^{-1}$) at the inlet. Note that this test case can thus be interpreted as a propagating wet-dry interface but, following Hunter et al. (2005) and Bates et al. (2010), we refer to it as a wave. In this test case, we evaluate the uncertainty associated with the spatially uniform Manning friction coefficient, $n_m$, and set this parameter to have a normal distribution $n_m \sim \mathcal{N}(0.03, 0.01)\,\mathrm{sm}^{-1/3}$. Note that, as Manning coefficients must be non-negative, any sampled values below 0 are discarded. We choose the Manning coefficient as our uncertain parameter because Bates et al. (2010) note that this test case is particularly sensitive to this parameter and thus this is a good test for our MLMF framework. The remaining parameters are the same as those in Leijnse (2018) and, in particular, we keep the simulated time at 1 h and the length in the $x$-direction equal to 5000 m. The quantity of interest is the expected value of the water elevation at the end of the simulation at $x = 1000$m, $x = 1500$m, $x = 2000$m and $x = 2500$m.

The advantage of this test case is that, due to the horizontal slope and the constant velocity condition at the inlet, the inviscid shallow water equations can be solved analytically with the following result

$$h(x, t) = \left( -\frac{7}{3} n_m^2 u^2 (x - ut) \right)^{3/7}, \tag{18}$$

where $h$ is the water level at any given location $x$ and time $t$, and $u$ the prescribed flow velocity at the boundary. The full analytical derivation can be found in Hunter et al. (2005), although in (18) we correct a typographical error in that work. Following Hunter et al. (2005) and Leijnse (2018), $u$ is set equal to 1 ms$^{-1}$. Using this analytical result, we can get a good estimate of the expected value of the true solution. However, we cannot find the 'true' expected value because of the uncertainty in $n_m$ and must instead run a Monte Carlo simulation varying $n_m$ in (18). Note that evaluating (18) is trivial and therefore the Monte Carlo simulation on the analytical result is very fast.

Before running the full MLMF algorithm, we run a small test using a spatially uniform Manning friction coefficient of $0.0364\,\mathrm{sm}^{-1/3}$ to compare the final water elevations from the SFINCS and XBeach models with the analytical result obtained from (18), and check they all approximately agree. We also check how the output variable varies with grid-size for both models. Figure 4 shows that the XBeach results agree more closely with the analytical result than SFINCS', which is to be expected as XBeach is the HF model. Nevertheless, the SFINCS results are not very different from the analytical result, indicating that it represents a good choice for the LF model. The effect of using a different resolution in both models is less clear in Figure 4, probably because there is both model error (here due to the numerical model possessing viscous dissipation while the analytical result is derived from the inviscid equations) and discretisation error (the error arising from using a finite mesh to solve the model equations). These two different types of error can, to some extent, cancel each other out if they have opposite signs.

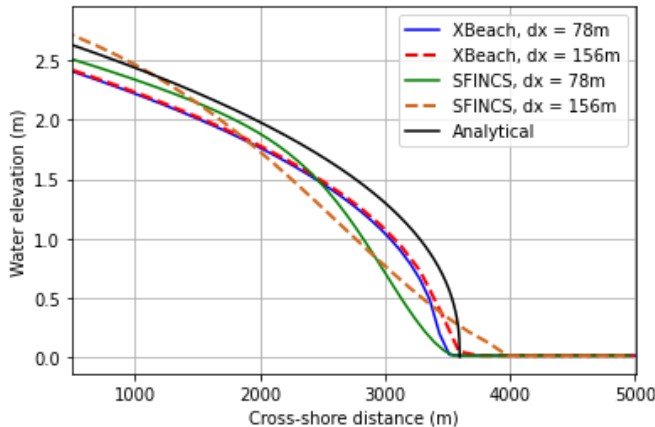

**Figure 4.** Comparing the final water elevation from using SFINCS and XBeach at $dx = 78$m and $dx = 156$m (corresponding to 64 and 32 elements in the $x$-direction respectively) with the analytical result for the non-breaking wave test case. A Manning friction coefficient of $0.0364\,\mathrm{sm}^{-1/3}$ is used in all simulations.

Hence in this example, the observed behaviour may be a consequence of the discretisation error decreasing whilst the model error stays the same as the resolution becomes finer, leading to an apparent increase in the total error.

For our MLMF simulation, we use grids with $2^l$ mesh cells in both SFINCS and XBeach, where the coarsest grid-size is $l = 4$ and the finest is $l = 10$, and consider $n_m \sim \mathcal{N}(0.03, 0.01)$. Table 1 compares the computational cost of running each of the models at these levels and shows that SFINCS is always much faster than XBeach. As the level number increases (*i.e.* the grid resolution becomes finer), unsurprisingly, the cost of both models increases, and, after level 8, this leads to the cost efficiency improvement from using SFINCS over XBeach increasing. The latter suggests that computational efficiency improvements will increase as the test case complexity increases and indicates that large computational cost improvements can be made by using MLMF. Before running the full MLMF algorithm (Algorithm 1), we first run Step 1 to determine key MLMF parameter values at each location. The left panel of Figure 5 shows that using the modified correlation in (9) means that SFINCS and XBeach are well correlated for almost every output location at every level, with almost perfect correlation at some locations. The worst correlation is at $x = 2500$ m, likely because SFINCS struggles with accurately simulating the front of the wave (see Figure 4). The impact of using this modified correlation formula is clearly shown in the right panel of Figure 5, which shows large increases in the modified correlation compared to the original correlation, especially at Level 6.

We can thus proceed to the next steps of the MLMF algorithm and compare our MLMF results to the analytical estimate (recall this is an estimate of the expected value of the true solution rather than the true expected value because of the uncertainty in $n_m$). We also compare our MLMF results with those obtained using the MLMC approach with SFINCS and XBeach separately. We initially use an accuracy tolerance of $\epsilon = 1 \times 10^{-3}$ in (14) and (B7) to calculate the optimum number of simulations for MLMF and MLMC, respectively. Note that, to calculate both the MLMC and MLMF estimators at level 10 (the finest level considered in this test case), we require simulations at the previous levels too (see Eqs. (B4) and (3)). Therefore we can truncate

| | Average time for single level run (s) | | Cost ratio | Grid resolution |
| | XBeach | SFINCS | $(\omega_l)$ | pair (m) |
|---|---|---|---|---|
| Level 4 | 1.72 | 0.0548 | 31 | $5000/(2^4,\ \ )$ |
| Level 5 | 3.85 | 0.138 | 28 | $5000/(2^5, 2^4)$ |
| Level 6 | 6.98 | 0.215 | 32 | $5000/(2^6, 2^5)$ |
| Level 7 | 11.8 | 0.383 | 31 | $5000/(2^7, 2^6)$ |
| Level 8 | 24.8 | 0.735 | 34 | $5000/(2^8, 2^7)$ |
| Level 9 | 62.2 | 1.44 | 43 | $5000/(2^9, 2^8)$ |
| Level 10 | 191 | 2.60 | 73 | $5000/(2^{10}, 2^9)$ |

**Table 1.** Summary of average time taken to run SFINCS and XBeach at each level for the non-breaking wave test case. As can be seen from Eqs. (B4) and (3), at every level (apart from the coarsest level) a pair of simulations at two different resolutions is required. These resolutions are shown in the 'Grid resolution pair' column and we recall that the same resolutions are used in each model.

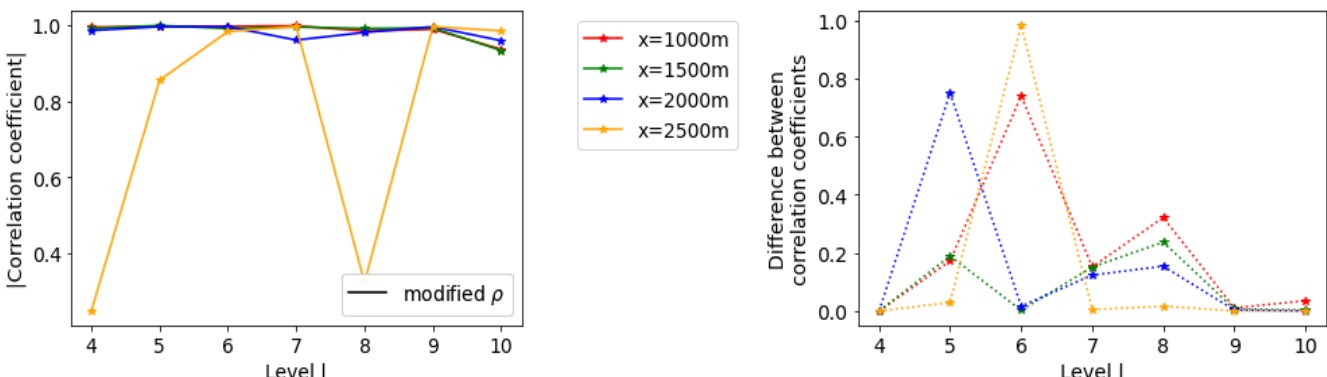

**Figure 5.** Comparing the real and modified correlation values between SFINCS and XBeach to find water elevation at specific locations in the non-breaking wave test case. Note that each colour represents a specific output location. Left: Absolute value of modified correlation (Eq. 9). Right: Difference between absolute value of modified and real correlation.

the MLMC/MLMF simulations and directly analyse how the error changes on the addition of each extra level. Figure 6 shows that, in general, the error in both the single model MLMC approaches and the MLMF approach with respect to the analytical estimate decreases as the grid resolution becomes finer. Furthermore, Figure 6a shows that the error from MLMC with XBeach and MLMF are very similar. In contrast, Figure 6b shows that MLMC with SFINCS is significantly less accurate than either MLMF or MLMC with XBeach, again justifying our choices of HF and LF models.

The error to the analytical estimate shown in Figure 6 includes both model error and discretisation error, but the main error component reduced by MLMC and MLMF is the discretisation error. Thus, we isolate the discretisation error by comparing the expected values of MLMF and MLMC to the expected values from using the standard Monte Carlo method with 500,000 simulations of XBeach at the finest resolution considered (1024 mesh cells in the $x$-direction). Note that to run this Monte

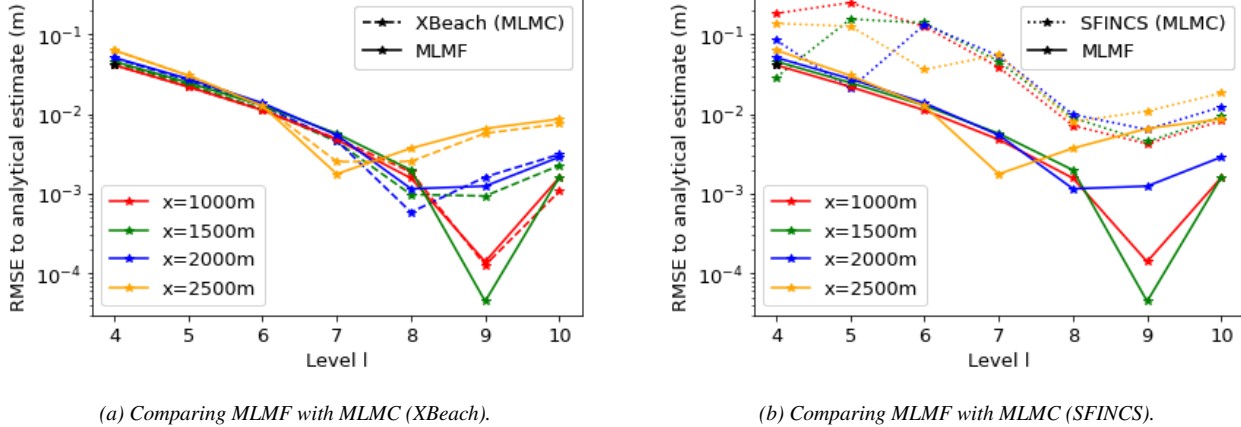

*(a) Comparing MLMF with MLMC (XBeach).*     *(b) Comparing MLMF with MLMC (SFINCS).*

**Figure 6.** Error (RMSE) with respect to the analytical estimate for the final water elevation at the locations of interest in the non-breaking wave test case, as the resolution level becomes finer. The $x$-axis indicates the finest level considered by the MLMC/MLMF estimator for that error, and both the error from using MLMF and the error from using MLMC with a single model are shown. Here a tolerance of $\epsilon = 1 \times 10^{-3}$ is chosen in (Eq. B7) and (Eq. 14) for MLMF and MLMC respectively.

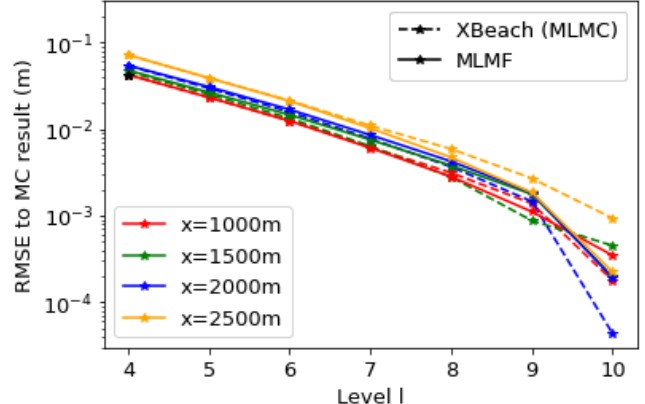

**Figure 7.** Error (RMSE) between the MLMF result and the Monte Carlo (MC) result for the analytical estimate for the final water elevation at the locations of interest in the non-breaking wave test case, as the resolution level becomes finer. The $x$-axis indicates the finest level considered by the MLMC/MLMF estimator for that error. Both the error from using MLMF and the error from using MLMC with a single model are shown. Here a tolerance of $\epsilon = 1 \times 10^{-3}$ is chosen in (Eq. 14) and (Eq. B7) for MLMF and MLMC, respectively.

Carlo simulation takes 1000 core days, or almost a month of wall clock time on the 40 core computer we had available. Figure 7 shows that, as the level becomes finer, the error to the Monte Carlo result, (*i.e.* the discretisation error) decreases uniformly for both MLMF and MLMC with XBeach, showing MLMF and MLMC are working correctly. Furthermore, as with the total error, the MLMF error and the XBeach (MLMC) error are very similar.

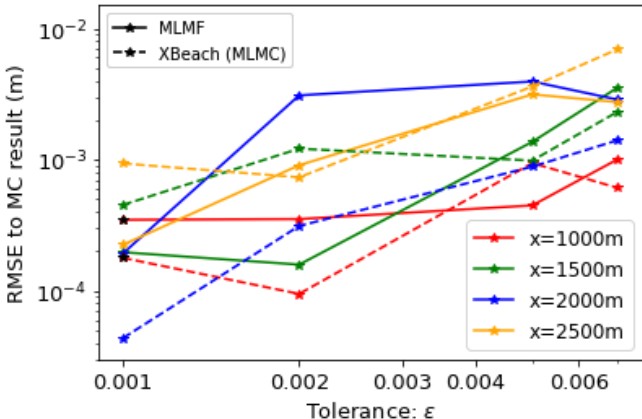

**Figure 8.** Error (RMSE) between the MLMF result and the Monte Carlo (MC) result as the tolerance value $\epsilon$ in (Eq. 14) is varied in the non-breaking wave test case. This is compared to the error (RMSE) when varying $\epsilon$ in (Eq. B7) when using MLMC with XBeach.

All the test case results shown so far in this section use the same accuracy tolerance of $\epsilon = 1 \times 10^{-3}$ in (14) and (B7). If MLMF is working as expected, the error in the MLMF result should decrease as the $\epsilon$ value decreases. Thus to verify this we re-run the test case using a range of tolerance values. Figure 8 shows that, indeed as $\epsilon$ decreases, both the error in the MLMF result and the error in the XBeach MLMC result decrease (with respect to the Monte Carlo result). More importantly,

the MLMF error is of the same order of magnitude as the XBeach MLMC error. Furthermore, Figure 9 shows that MLMF achieves this accuracy using significantly fewer HF simulations, with generally a difference of one order of magnitude. To do so, MLMF also requires $(r_l+1)N_l^{HF}$ LF simulations. Figure 10 shows that $r_l$ is approximately 10 at all levels for this test case (*i.e.* $\mathcal{O}(10)$ times more LF simulations are required than HF simulations), a small factor given the computational cost savings shown in Table 1 from using SFINCS. Figure 9 also shows that the optimum number of XBeach simulations required for both

MLMC and MLMF does not decrease uniformly, but instead increases at level 10 relative to the coarser level 9 for large $\epsilon$. However, the number of simulations is so small (less than 10 and also less than the total number of processing cores used) that this does not make a significant difference to the computational cost. Thus, overall, Figure 9 and 10 suggest that notable computational cost savings can be made in this test case by using MLMF.

As discussed in Section 2.2, we can use the modified inverse transform sampling method (17) to also generate the cumulative

distribution function (CDF) from the MLMF outputs (here produced using $\epsilon = 1 \times 10^{-3}$). These can then be used to readily assess the likelihood of a certain high water level occurring and greatly improve our understanding of the test case. Figure 11 demonstrates how the CDF generated in this manner agrees qualitatively with the CDF generated using Monte Carlo outputs. Using this figure we can determine, for example, that the elevation height at $x = 2500$m is very certain but at $x = 1000$m there is an almost equal probability that it could be less than 1.5 m or more than 2.5 m. In physical terms, this means that

friction is important for determining the slope of the final water level close to the boundary, but the final wave-front shape is more stable and less affected by friction. The agreement between the MLMF and Monte Carlo CDFs is quantified in Table 2

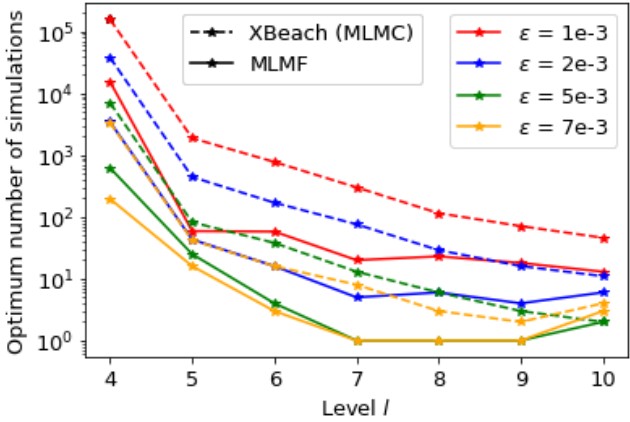

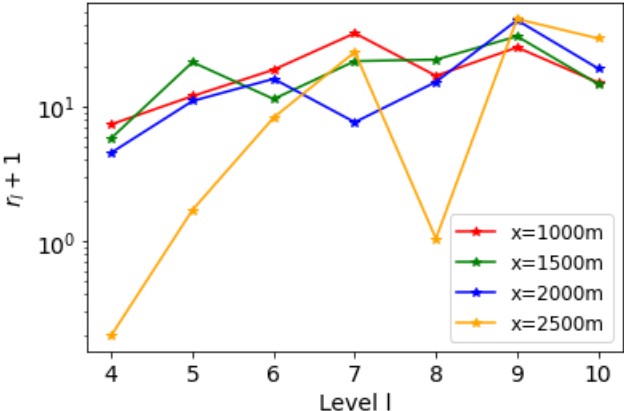

**Figure 9.** Optimum number of XBeach (HF) simulations required by MLMF (Eq. 14) and MLMC (Eq. B7) for the non-breaking wave test case. The number required by MLMF is always substantially fewer than that required by MLMC.

**Figure 10.** Factor of total LF simulations ($r_l + 1$) required by MLMF compared to number of HF simulations for the non-breaking wave test case, where $r_l$ is (Eq. 13).

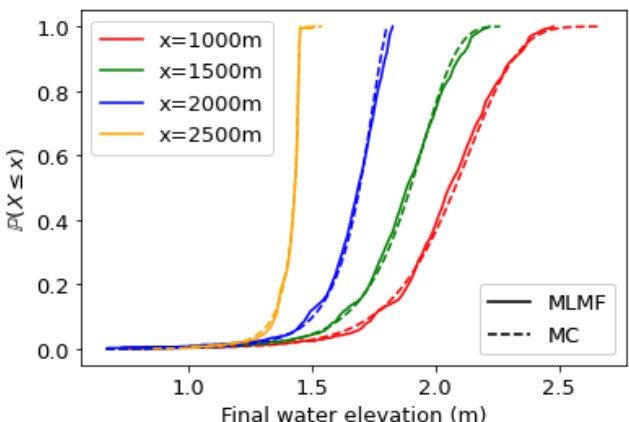

**Figure 11.** CDFs generated from MLMF outputs using the modified inverse transform sampling method (Eq. 17) compared with those generated using Monte Carlo (MC) outputs for the non-breaking wave test case.

where we calculate the $L^2$ error norm and maximum error norm between them. Note that we evaluate the CDFs at 100 equally spaced points and, therefore, the implementation of the $L^2$ error norm is equivalent to calculating the RMSE between the two CDFs. The small error norms in Table 2 give us confidence in our new modified inverse transform sampling method's ability to accurately generate CDFs.

Finally, throughout this section, we have assumed that either we can approximate the expected value of the true solution or we can approximate the expected value of the XBeach simulation by using Monte Carlo with large numbers of simulations at fine

**Table 2.** $L^2$ error norm and maximum error norm between the MLMF and Monte Carlo CDFs for the non-breaking wave test cases. Note the error norms are unitless because the CDFs are unitless.

| Test Case | $L^2$ **Error Norm** | **Max. Error Norm** |
|---|---|---|
| Non-breaking wave test case – 1000 m | $1.5 \times 10^{-2}$ | $4.5 \times 10^{-2}$ |
| Non-breaking wave test case – 1500 m | $1.4 \times 10^{-2}$ | $4.0 \times 10^{-2}$ |
| Non-breaking wave test case – 2000 m | $1.6 \times 10^{-2}$ | $7.1 \times 10^{-2}$ |
| Non-breaking wave test case – 2500m | $9.3 \times 10^{-3}$ | $3.9 \times 10^{-2}$ |

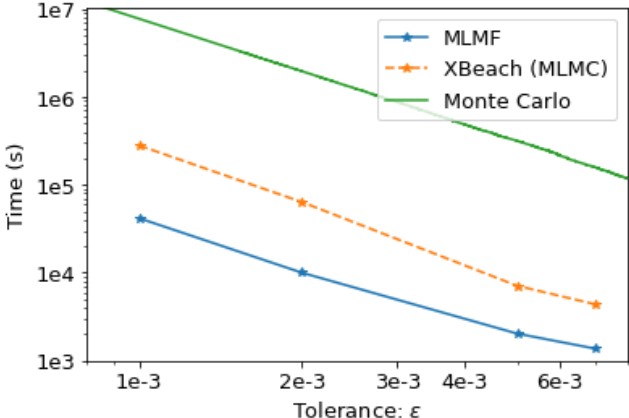

**Figure 12.** Comparing the computational cost required to achieve tolerance $\epsilon$ using MLMF, XBeach with MLMC and the Monte Carlo method for the non-breaking wave test case.

resolutions. However, if MLMF is to be of use, we need to apply it to cases where the 'true' value is not known. In these cases, the only parameter the practitioner can use to check accuracy is the tolerance value $\epsilon$ in the constraint (12), which we recall

here is $\mathrm{Var}\left[\hat{Q}_{M_L}^{HF,CV}\right] < \epsilon^2/2$. Figure 12 compares the computational cost required by MLMF, MLMC and the Monte Carlo method to satisfy this constraint. MLMC and MLMF ensure this through the formulae for the optimum number of simulations and, thus, for these methods, $\epsilon$ is plotted against the computational cost of the optimum number of simulations used. For the Monte Carlo method, the value of $\epsilon$ is equal to the square root of twice the variance calculated after each simulation, and is plotted against the time taken to run that number of simulations. This is an imperfect measure of accuracy for Monte Carlo, but

the best available to us. Figure 12 shows that, even for this simple test case, MLMF is more than a hundred times faster than Monte Carlo, and on average, five times faster than MLMC combined with XBeach alone. This is a very promising result for such a simple 1D test case and suggests that MLMF represents a good method for improving computational efficiency, whilst still achieving accurate results.

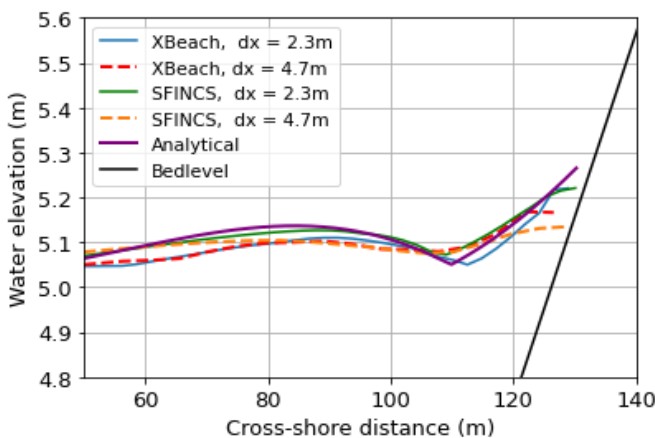

**Figure 13.** Comparing the maximum elevation height achieved at every point in the domain over the entire simulation when using SFINCS and XBeach at $dx = 2.3$m and $dx = 4.7$m (corresponding to 64 and 32 mesh cells in the $x$-direction respectively) with the analytical result for the Carrier-Greenspan test case. A slope of 0.041 is used in all simulations. Note that the water elevation height does not always meet the bedlevel because the bedlevel is slightly differently defined at the different resolutions.

### 3.2 Carrier-Greenspan test case

For our second test case, we consider the 1D Carrier-Greenspan test case, first introduced in Carrier and Greenspan (1958), where a harmonic, non-breaking infragravity wave travels over a plane sloping frictionless beach. This test case is more complex than our first case because it requires the simulation of run-up and run-down, but Leijnse (2018) and Leijnse et al. (2021) show it can be successfully simulated using both SFINCS and XBeach. Following these works, we generate a wave train using a varying elevation boundary condition at the inlet, which results in a wave period of 48 s.

In this section, we evaluate the uncertainty associated with the linear bedslope and assume it has a normal distribution, slope $\sim \mathcal{N}(0.04, 0.02)$. Note that any samples below 0.005 (i.e. slope 1:200) are discarded because otherwise the domain is completely wet. We choose the slope as our uncertain parameter because it represents a significant source of uncertainty, as discussed in Unguendoli (2018), particularly when simulating run-up and run-down as is the case here. The remaining parameters are the same as those used in Leijnse (2018), in particular, the length in the $x$-direction is 150 m and the simulated period is 384 s. An advantage of the Carrier-Greenspan test case is that there exists an analytical result. A full derivation of the analytical result used in our work is given in Carrier and Greenspan (1958) and is based on solving the dimensionless inviscid shallow water equations where friction is ignored.

When a wave runs up a slope, often the quantity of most interest is not the water depth at a particular location in time but, instead, the (maximum) run-up height. Thus, for this test case, our quantity of interest is the maximum run-up height over the whole simulated period. Here we take the run-up height to be the water elevation above a fixed datum in the last wet cell in the domain (water depths higher than 0.005 m). We first test how the maximum elevation height over the whole domain depends

|  | Average time for single level run (s) | | Cost ratio | Grid resolution |
|---|---|---|---|---|
|  | XBeach | SFINCS | $(\omega_l)$ | pair (m) |
| Level 5 | 2.86 | 0.66 | 4 | $150/(2^5, \quad)$ |
| Level 6 | 8.57 | 1.17 | 7 | $150/(2^6, 2^5)$ |
| Level 7 | 17.3 | 1.12 | 15 | $150/(2^7, 2^6)$ |
| Level 8 | 35.4 | 1.15 | 31 | $150/(2^8, 2^7)$ |
| Level 9 | 76.6 | 1.33 | 58 | $150/(2^9, 2^8)$ |

**Table 3.** Summary of average time taken to run SFINCS and XBeach at each level for the Carrier-Greenspan test case. As can be seen from Eqs. (B4) and (3), at every level (apart from the coarsest level) a pair of simulations at two different resolutions is required. These resolutions are shown in the 'Grid resolution pair' column and we recall that the same resolutions are used in each model.

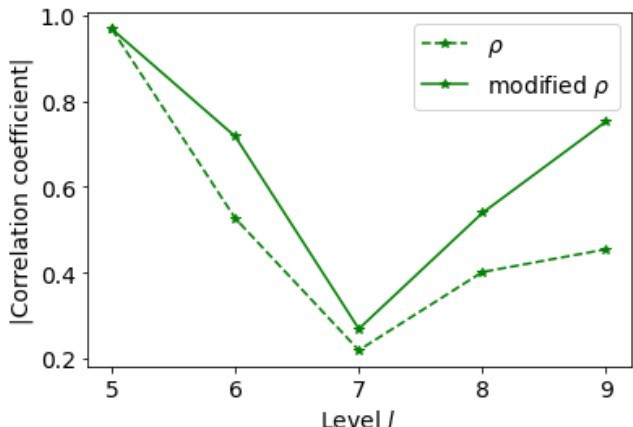

**Figure 14.** Comparing the real and modified correlation values between SFINCS and XBeach to determine the maximum run-up height in the Carrier-Greenspan test case.

on the resolution and model used in the simulation. Figure 13 shows that both models at both resolutions underpredict the run-up height relative to the analytical result and that, whilst the XBeach high resolution result is the most accurate and the SFINCS low resolution result the least accurate, the high resolution result using SFINCS is better than the low resolution result
of XBeach, which is a promising outcome.

For our MLMF simulation, we use grids with $2^l$ mesh cells in both SFINCS and XBeach, where the coarsest grid-size is $l = 5$ and the finest is $l = 9$. Table 3 compares the computational cost of running each of the models at these levels and shows that, as with the previous test case, SFINCS is always much faster than XBeach. As the resolution becomes finer, the computational cost of XBeach increases but the computational cost of SFINCS remains relatively constant. We hypothesise this is because
there is always a start-up cost to begin the SFINCS model run and this dominates the overall cost for such a small test case with a short runtime. As with the previous test case, the cost ratio between SFINCS and XBeach increases as the resolution becomes finer, which is again a promising indication of the efficiency gains we can expect from using MLMF. Before running

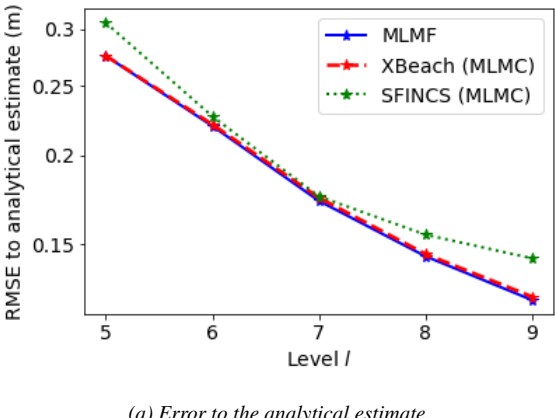

(a) Error to the analytical estimate.

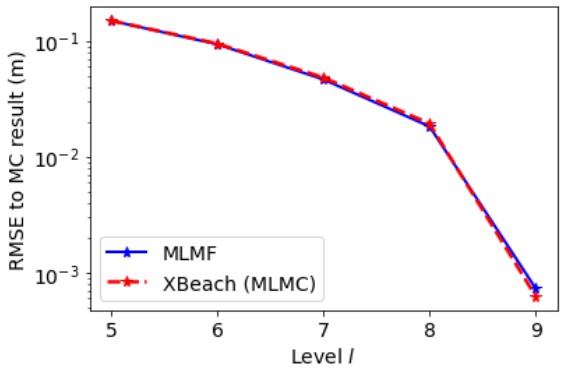

(b) Error to the Monte Carlo result.

**Figure 15.** Error (RMSE) in the maximum run-up height as the resolution level becomes finer for the Carrier-Greenspan test case. The $x$-axis indicates the finest level considered by the MLMC/MLMF estimator for that error and the errors from using MLMF and from using MLMC with a single model are both shown. A tolerance of $\epsilon = 1 \times 10^{-3}$ is used in (Eq. 14) and (Eq. B7) for MLMF and MLMC, respectively.

the full MLMF algorithm (Algorithm 1), we determine the values of key MLMF parameters using Step 1. The correlation is of particular interest and Figure 14 shows that using the modified correlation formula (9) leads to increased correlation between
the two models at all levels, although this increase is small.

Running the next steps in the MLMF algorithm, we can compare our MLMF results to the analytical estimate and to the Monte Carlo result estimated using 400,000 simulations of XBeach at the finest resolution (512 mesh cells in the $x$-direction) which takes almost 400 days of core time to run. Note that, as in the previous test case, due to the uncertainty in the slope, the analytical estimate is not the 'true' expected value, but instead an estimate of the expected value of the true solution. We also
run the MLMC algorithm with SFINCS and XBeach separately. Note that we initially choose a tolerance of $\epsilon = 1 \times 10^{-3}$ in (14) and (B7) for MLMF and MLMC, respectively. As in the previous test case, we can truncate the MLMC/MLMF simulations at intermediate levels and directly analyse how the error changes on the addition of each extra level. Figure 15a shows that the error to the analytical estimate decreases uniformly for all methods indicating that the error could be further decreased by using finer levels of resolution. Furthermore the figure shows that the error using MLMF and MLMC with XBeach is lower than that
using SFINCS. This justifies that XBeach is the HF model for this test case and that the MLMF approach can achieve the same, or lower, error than using only the HF model. Furthermore, Figure 15b shows a similar trend for the error to the Monte Carlo result, with the error decreasing uniformly for all methods, and having a similar value for MLMF and MLMC with XBeach.

So far in this section we have only considered a single tolerance value. Therefore, we re-run this test case using different tolerance values $\epsilon$ in (14) and (B7) for MLMF and MLMC, respectively. Figure 16 shows that the MLMF and XBeach MLMC
errors decrease as the tolerance decreases. Most importantly, this figure shows that MLMF is approximately as accurate as using MLMC with XBeach. Additionally, Figure 17 shows that the optimum number of HF simulations required by the MLMF algorithm to achieve this accuracy is always less than that required by MLMC. To achieve this, MLMF also requires $(r_l + 1)N_l^{HF}$ LF simulations and Figure 18 shows that $r_l$ is less than 10 at all levels for this test case. Furthermore, at level 7 (and,

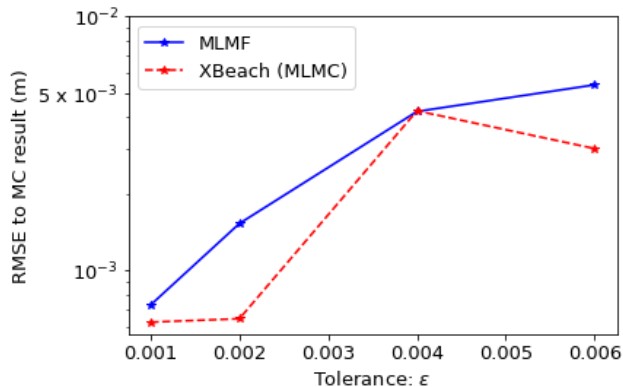

**Figure 16.** Error (RMSE) between the MLMF result and the Monte Carlo (MC) result for the Carrier-Greenspan test case as the tolerance value $\epsilon$ in (Eq. 14) is varied. This is compared to the error when varying $\epsilon$ in (Eq. B7) for MLMC with XBeach.

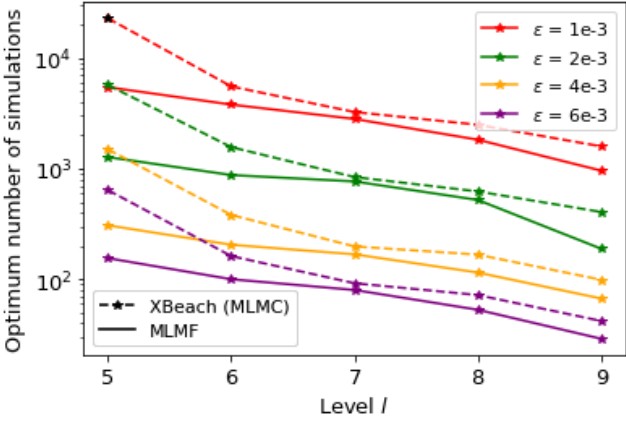

**Figure 17.** Optimum number of XBeach (HF) simulations required by MLMF (Eq. 14) and MLMC (Eq. B7) for the Carrier-Greenspan test case. The number required by MLMF is always substantially fewer than that required by MLMC.

**Figure 18.** Factor of total LF simulations $(r_l + 1)$ required by MLMF compared to number of HF simulations for the Carrier-Greenspan test case, where $r_l$ is (Eq. 13).

to a lesser extent, level 8), the difference between the optimum number of simulations required is smaller than at other levels
because the correlation between SFINCS and XBeach is lower (see Figure 14), meaning MLMF and MLMC with XBeach are almost equivalent. This is also reflected in a lower factor of total LF simulations in Figure 18. This highlights the importance of choosing two closely correlated models to ensure optimum efficiency from using the MLMF method.

As in the previous test case, we can apply the modified inverse transform sampling method to the MLMF output (from using $\epsilon = 1 \times 10^{-3}$) to generate a CDF. This CDF can be used to readily assess flooding potential, for example, Figure 19 shows
that the probability the run-up height will exceed 5.48 m is 5%. This information can then be used, for example, to inform

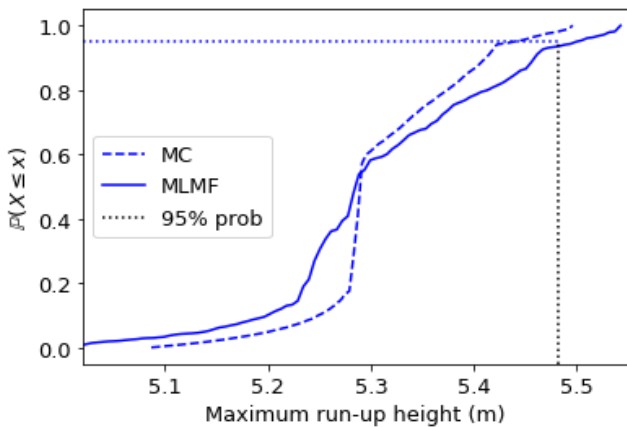

**Figure 19.** CDFs generated from MLMF outputs using the modified inverse transform sampling method (Eq. 17) compared with those generated using Monte Carlo (MC) outputs, for the Carrier-Greenspan test case.

**Table 4.** $L^2$ error norm and maximum error norm between the MLMF and Monte Carlo CDFs for the Carrier-Greenspan test cases. Note the error norms are unitless because the CDFs are unitless.

| Test Case | $L^2$ **Error Norm** | **Max. Error Norm** |
|---|---|---|
| Carrier-Greenspan test case | $2.7 \times 10^{-2}$ | $2.3 \times 10^{-1}$ |

a local authority that it would be unwise to place a permanent building structure below this height, but a temporary beach structure might be ok. Figure 19 also shows that the CDF generated using MLMF outputs agrees fairly well with the Monte Carlo-generated CDF. We have quantified the agreement between the two CDFs in Table 4, where we calculate the $L^2$ error norm and maximum error norm between them. The maximum error norm is larger here than for the previous test case because 420  MLMF struggles to represent the steep change in the CDF at around 5.3 m. However, the $L^2$ error norm, which we recall is equivalent to the RMSE between the two CDFs, is small and indicates that overall the MLMF-generated CDF represents a good approximation and gives further confidence in our ability to accurately generate CDFs from MLMF outputs.

Finally, as discussed in the previous test case, in reality, the 'true' value of the quantity of interest is not always known and the only parameter available to check accuracy is the tolerance value $\epsilon$. Figure 20 compares the computational cost required 425  by MLMF, MLMC and the Monte Carlo method to satisfy the constraint (12) which we recall here is $\mathrm{Var}\left[\hat{Q}_{M_L}^{HF,CV}\right] < \epsilon^2/2$. As before, for MLMC and MLMF, $\epsilon$ is the tolerance and is plotted against the cost required to run the optimum number of simulations for this $\epsilon$. However, for Monte Carlo, $\epsilon$ is the square root of twice the variance calculated after each simulation and is plotted against the time taken to run that number of simulations. Thus, Figure 20 shows that using MLMF is at least three times as efficient as the Monte Carlo method for the same tolerance, and using MLMC is at least twice as efficient. Whilst these 430  improvements from using MLMF versus MLMC are not as notable as for the previous test case, they nevertheless show that

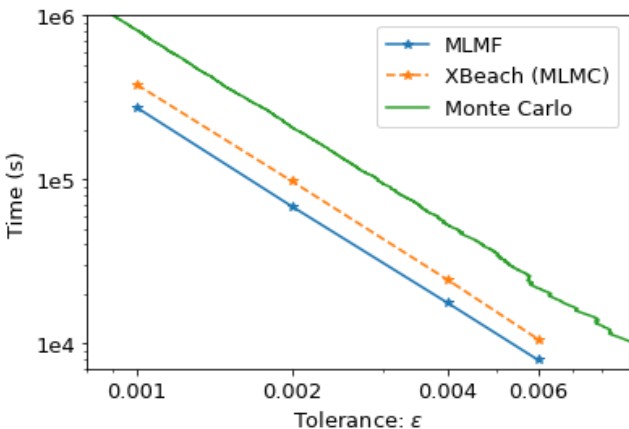

**Figure 20.** Comparing the computational cost required to achieve tolerance $\epsilon$ using MLMF, XBeach with MLMC and the Monte Carlo method for the Carrier-Greenspan test case.

even small differences between the number of optimum HF simulations (see Figure 17) are sufficient for MLMF to be more efficient than MLMC.

### 3.3 Myrtle Beach

The test cases considered so far in this work have been relatively simple one-dimensional idealised test cases. For our final test case, we consider the real-world test case of a dune system near Myrtle Beach, South Carolina, USA (see Figure 21). The bedlevel data of the specific beach of interest is shown in Figure 22. The goal of this test case is to estimate the maximum water depth (with respect to the bed level) due to flooding at various locations. Over the coming decades, climate change will lead to changing water levels but the actual change at specific locations is uncertain, which in turn leads to uncertainty in the impact of flooding from future storms. Thus, in this test case, we consider the offshore water level to be uncertain.

As in the previous test cases, we use XBeach as the HF model and SFINCS as the LF model in the MLMF algorithm. To simulate the waves in XBeach we use the surfbeat model mode with the JONSWAP wave spectrum (Hasselmann et al., 1973) and set the significant wave height equal to 4 m, the peak wave period equal to 12 s, the peak enhancement factor (used to alter the spectrum for fetch-limited oceans) equal to 3.3 and the main wave angle perpendicular to the shore equal to $124.3°$. Note that, as discussed in Section 1, SFINCS does not explicitly simulate short-waves (representing here a simplification in the setup of the LF model) and therefore we do not have to define a wave spectrum for it. In order to accurately model waves in the HF model of XBeach, we need a long stretch of water before the waves reach the beach, which is not present in the domain in Figure 22. Therefore, we extend the domain offshore, as shown in Figure 23, when running XBeach but, for reasons of computational cost, use the original smaller domain in SFINCS. For the larger XBeach domain, we maintain a uniform grid spacing in the original domain region (*i.e.* the region where both SFINCS and XBeach are simulated) so that the grids in each model are the same in that region. In the extended part of the domain, however, we vary the cross-shore grid resolution





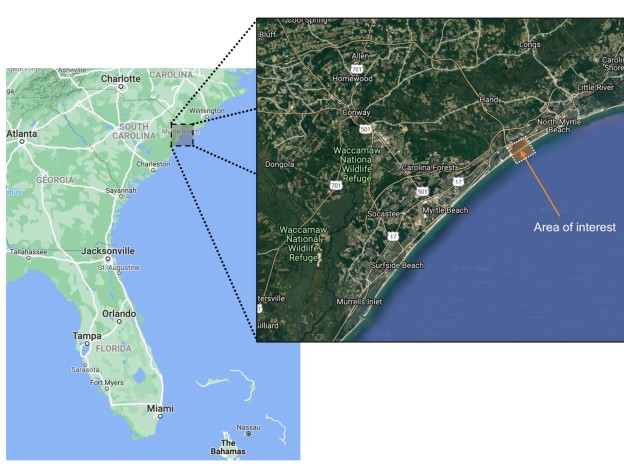

**Figure 21.** Location of area of interest in the Myrtle Beach test case. Source: © Google Maps 2021.

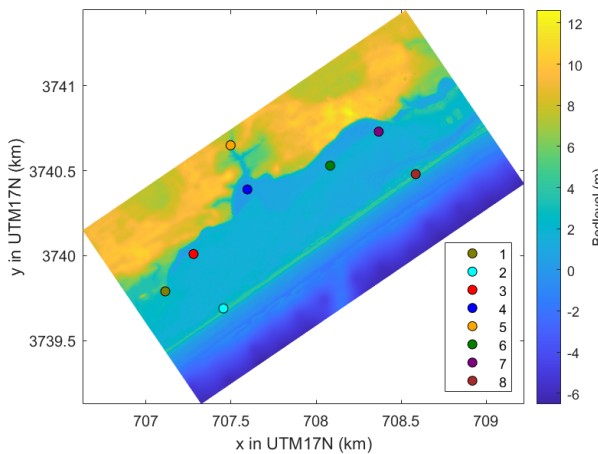

**Figure 22.** Bedlevel data for original non-extended domain of the Myrtle Beach test case with locations of interest marked with a circle. The locations are colour-coded and these colours are used to represent them throughout this section. Note that $x$ and $y$ are the Universal Transverse Mercator (UTM) co-ordinates for the global zone that Myrtle Beach is located in (17N).

depending on the bedlevel so as to make XBeach more computationally efficient (see Figure 23). The original non-extended domain is [0, 1000]m in Figure 23 and the extended part stretches from [-5250, 0]m. Note that we use the cross-shore grid size of the original non-extended domain as a lower bound for the grid-size in the extended domain. Therefore the grid in the extended domain also varies at each level, so as to make the cost comparisons between the levels fair. Finally, the grid-size parallel to the shore is kept constant (10 m) for simplicity and because in this test case we are most interested in cross-shore changes.

As this is a real-world study, we must also consider tides. These tides can have a large impact on coastal flooding and thus, for this test case, we evaluate the uncertainty in the maximum tide height, $h_{\text{tide}}$. In both SFINCS and XBeach, tides are modelled using a varying elevation height boundary condition at the offshore open boundary and in this test case this boundary condition follows

$$
\text{tide}(t) = \begin{cases} \frac{h_{\text{tide}}}{3600}t & 0\text{s} \leq t \leq 3600\text{s}, \\ h_{\text{tide}} & 3600\text{s} < t \leq 7200\text{s}, \\ -\frac{h_{\text{tide}}}{3600}(t - 10800) & 7200\text{s} < t \leq 10800\text{s}, \end{cases} \tag{19}
$$

which approximates a slightly sped up tidal signal relative to real-world tides, for reasons of computational cost. Note that we run the simulation for 3 h (10800 s). Due to the presence of a wave component in XBeach, we expect it is simulate overtopping significantly more accurately than SFINCS. To check whether MLMF still holds for problems with overtopping, we assume

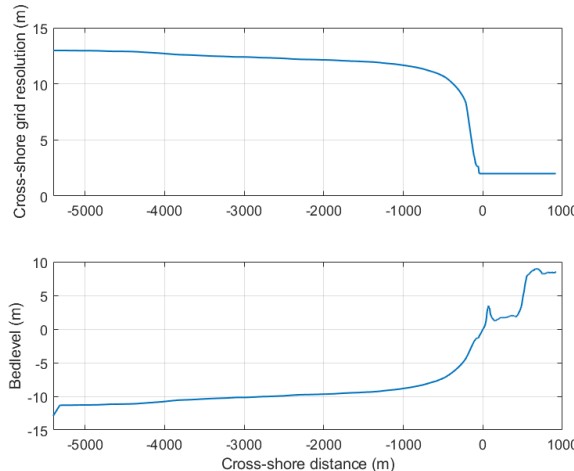

**Figure 23.** Varying cross-shore grid resolution based on the bedlevel in the extended XBeach domain. In the original non-extended domain, the cross-shore resolution is 2 m and thus a maximum resolution of 2 m is used in the extended domain. For illustration purposes, we show the cross-shore cross-section at the left-hand side edge of the domain.

that the maximum tide height has the distribution $h_{\text{tide}} \sim \mathcal{N}(5, 0.75)$m, so as to ensure overtopping of the first row of dunes which are approximately 4 m high (see Figure 23). The quantity of interest is then the maximum water depth at eight different locations in the domain, which are marked with coloured circles in Figure 22. Note that these colours are used to identify these locations in all figures throughout this section. Figure 24 shows an example of the maximum elevation height (relative to a fixed datum) computed by an XBeach simulation for this test case overlaid on a satellite image of the beach. The figure shows

that for this particular value of $h_{\text{tide}}$ (4.97 m), a substantial amount of overtopping occurs.

With the set-up of the test case complete, we now consider the MLMF set-up. For the MLMF simulation, we use grids with $\lceil \psi \times 2^l \rceil$ mesh cells in the cross-shore direction in the original non-extended domain, where $\psi$ is a user-defined factor here set equal to $155/4$. The coarsest grid-size is $l = 1$ and finest grid-size is $l = 4$. As the cross-shore distance in the original non-extended domain is 1240 m, this means the coarsest cross-shore resolution is 16 m and the finest one is 2 m. Note that,

throughout this test case, the resolution parallel to the shore is kept constant (10 m) as discussed above. As a first test, we compare how the values of the variable of interest depend on which model is used and the grid-resolution. In order to be able to distinguish between the model results at different locations, Figure 25 shows the maximum water depth (*i.e.* maximum water elevation height minus bedlevel) rather than the maximum water elevation height, which is our variable of interest. It shows that SFINCS results in lower predictions of the maximum water depth compared to XBeach due to the former omitting wave-driven

processes and, to a lesser extent, that the coarser resolution also results in an underprediction in both SFINCS and XBeach. The difference between the SFINCS and XBeach results is roughly the same at all locations. This simple shift is promising as it means that the models are likely to be correlated and MLMF can just adjust for the shift in predicted values. Furthermore, the maximum water depth in SFINCS and XBeach follows the same pattern between locations, which is a promising result.

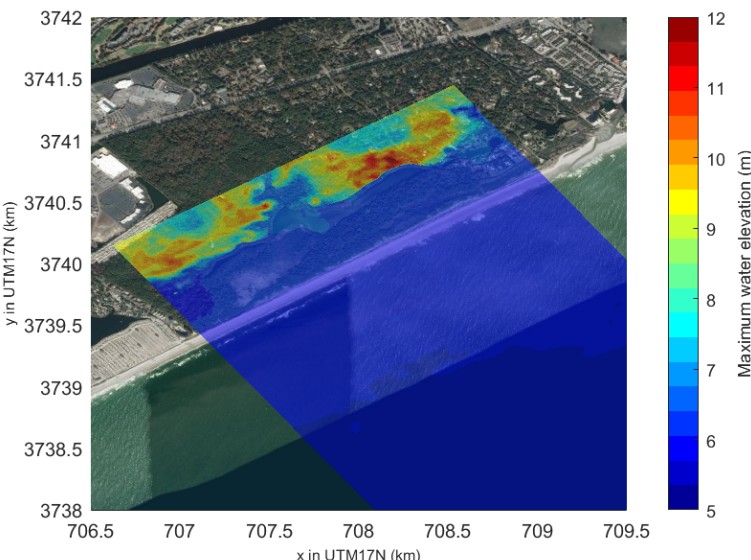

**Figure 24.** Maximum elevation height (relative to a fixed datum) from an example XBeach simulation for the Myrtle Beach test case, showing overtopping. This has been simulated using the grid resolution from Figure 23 and $h_{\text{tide}} = 4.97$m. The maximum elevation height has been overlaid on a satellite image of the location, to highlight the impact of coastal features on the elevation height.

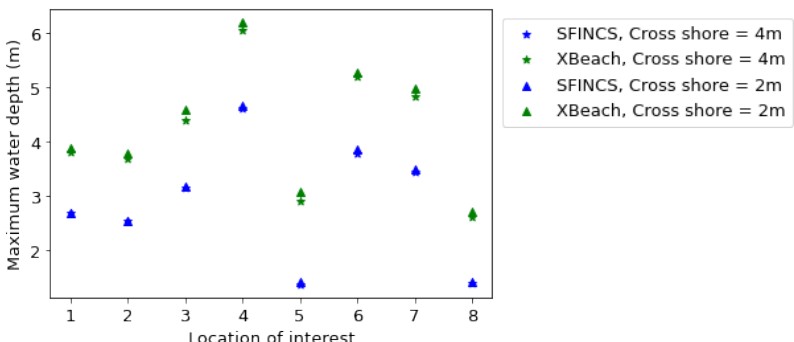

**Figure 25.** Comparing the maximum water depth achieved at the eight locations of interest using SFINCS and XBeach at two different resolutions, for the Myrtle Beach test case. A maximum tide height of $h_{\text{tide}} = 4.97$m is used in all simulations.

Table 5 compares the computational cost of running each of the models at the levels considered and shows that, as with the previous test cases, SFINCS is substantially faster than XBeach. Unlike with the other test cases, the cost ratio between the two models decreases as the resolution becomes finer. However, for this test case, the cost ratio is so large that even with this decrease, SFINCS is still 400 times faster than XBeach at the finest level. This indicates that substantial computational savings can be made by using a multifidelity approach in this complex real-world test case.

|  | Average time for single level run (s) | | Cost ratio | Grid resolution |
|  | XBeach | SFINCS | $(\omega_l)$ | pair (m) |
| --- | --- | --- | --- | --- |
| Level 1 | 31,926 | 14 | 2236 | $32(2^1, \quad)$ |
| Level 2 | 80,115 | 45 | 1769 | $32/(2^2, 2^1)$ |
| Level 3 | 155,033 | 178 | 872 | $32/(2^3, 2^2)$ |
| Level 4 | 388,664 | 963 | 403 | $32/(2^4, 2^3)$ |

**Table 5.** Summary of average time taken to run SFINCS and XBeach at each level for the Myrtle Beach test case. As can be seen from Eqs. (B4) and (3), at every level (apart from the coarsest level) a pair of simulations at two different resolutions is required. These resolutions are shown in the 'Grid resolution pair' column and we recall that the same resolutions are used in each model.

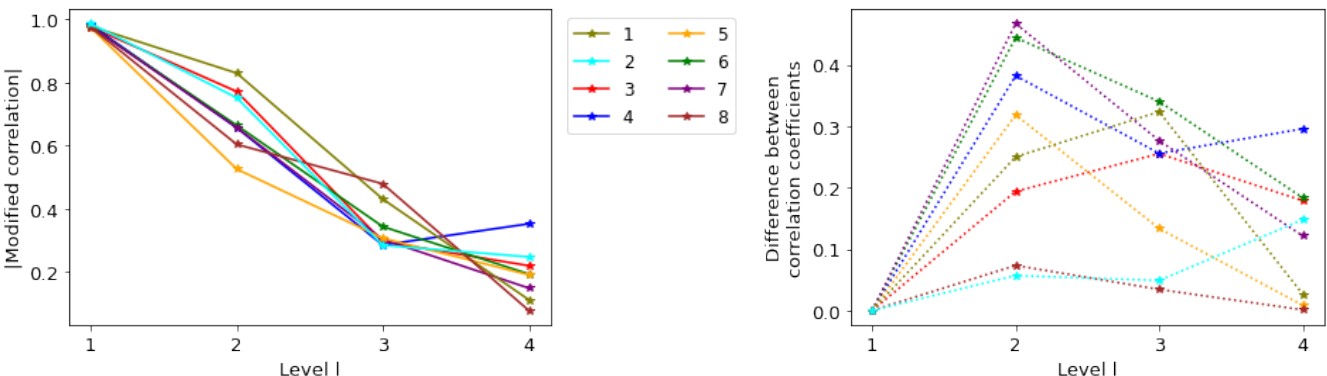

**Figure 26.** Comparing the real and modified correlation values between SFINCS and XBeach to determine maximum water elevation at eight specific locations in the Myrtle Beach test case. Left: Absolute value of modified correlation (Eq. 9). Right: Difference between absolute value of modified and real correlation.

As with the previous test cases, before running the full MLMF algorithm, we first analyse the values of key MLMF param-
eters determined in Step 1 of the algorithm (Algorithm 1). Figure 26 shows that the modified correlation (9) between SFINCS and XBeach generally decreases as the resolution level increases and that, at the finest level, the correlation is very low at some locations. The right panel of Figure 26 shows that the modified correlation method is very beneficial in this test case because it results in a large increase in correlation, especially at level 2. However, at the finest levels, there is almost no increase in the correlation at several locations. The conclusion from Figure 26 is therefore that, unsurprisingly, the benefits of the more complete and complex physics implemented in XBeach become greater as the mesh becomes finer.

Before running the full MLMF algorithm, we also consider how to assess the accuracy of the MLMF algorithm for this test case. This is a complex computationally expensive real-world problem for which there is no analytical solution and for which approximating a 'true' solution using the standard Monte Carlo method at the finest resolution considered is impractical (each simulation of XBeach at this resolution takes on average 3 days). Therefore, to assess accuracy, we use the following general

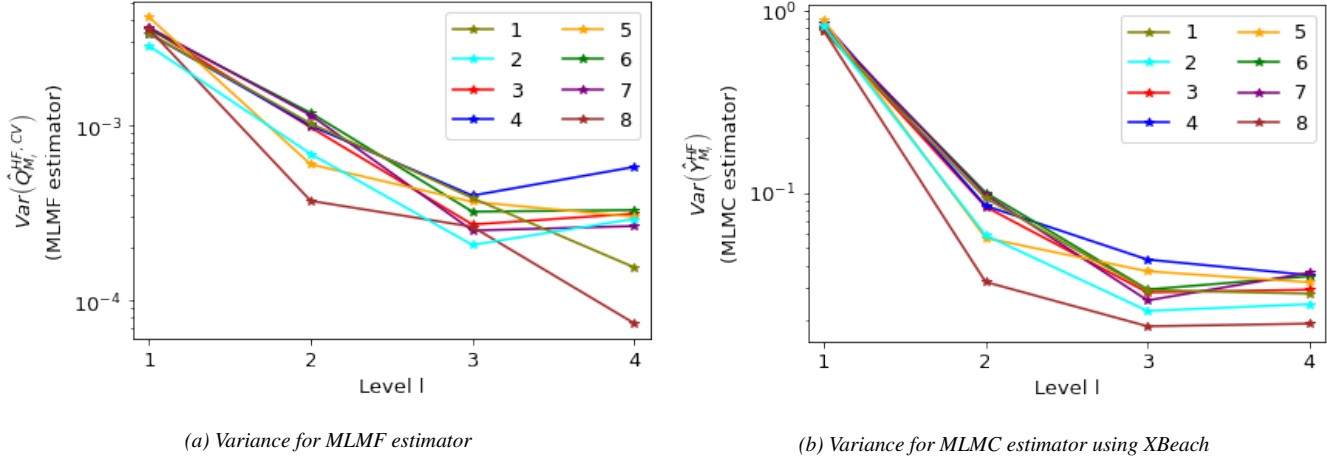

*(a) Variance for MLMF estimator*

*(b) Variance for MLMC estimator using XBeach*

**Figure 27.** Comparing the different rates at which variance of the MLMF estimator (Eq. 6) and the variance of the MLMC estimator for XBeach (Eq. B5) decrease as the resolution becomes finer. The $x$-axis indicates the finest level considered by the MLMC/MLMF estimator. Note the MLMF variance is two orders of magnitude smaller than the MLMC variance.

theoretical formula for the root mean squared error (RMSE)

$$\text{RMSE} = \sqrt{\mathbb{E}[(\hat{Y} - \mathbb{E}[X_L])^2] + (\mathbb{E}[X_L] - \mathbb{E}[X])^2}, \tag{20}$$

where $X$ is the true solution, $X_L$ is the solution on the finest level (*i.e.* level $L$), and $\hat{Y}$ can be either the MLMF estimator $\hat{Q}_{M_l}^{HF}$ or the MLMC estimator $\hat{Y}_{M_l}^{HF}$. The first term in (20) is the only term affected by whether MLMC or MLMF is used. Therefore given that the purpose of this work is to verify MLMF, it is not important that the true solution $X$ is unknown. Instead, for 505   this test case, we use the the first term (the estimator variance) as a proxy for the RMSE and estimate it using the output generated in Step 1 of the MLMC and MLMF algorithms. As in the previous test cases, we can truncate the MLMC/MLMF simulations before the finest level and thus directly analyse how the variance changes on the addition of each extra level. Figure 27 shows how both the MLMF variance (6) and MLMC variance (B5) vary with level $l$. For both estimators at all locations, the general trend is that the variance decreases as the resolution level increases. This is an important result because it means 510   that fewer simulations are required on the finer levels. The MLMC variance, however, plateaus and then increases slightly at some locations for the finer resolutions, whereas the decrease in MLMF is more uniform, indicating that MLMF is performing better than MLMC for this test case. More importantly, the variance of the MLMF estimator is two orders of magnitude smaller than that of the MLMC estimator. Thus, using the RMSE formula (20), MLMF is more accurate than MLMC, although this is difficult to determine without an approximation to the 'true' solution. The smaller variance also means that MLMF will require 515   fewer HF simulations than MLMC and, therefore, be more computationally efficient.

Given these promising results, we can now run the full MLMF algorithm (Algorithm 1) choosing a tolerance of $\epsilon = 3 \times 10^{-2}$ in (14). Figure 28 shows the spatial representation of the final expected value estimated using MLMF at the locations of interest. It shows that the expected maximum elevation height grows as we move inland, especially as the water gets funneled into the

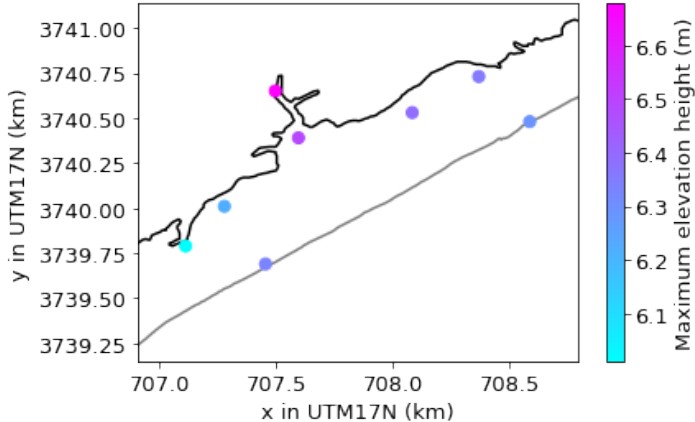

**Figure 28.** Spatial representation of the expected value of the maximum elevation height estimated using MLMF with a tolerance of $\epsilon = 3 \times 10^{-2}$ in (Eq. 14).

inlet. This is a physically realistic result and therefore gives us further confidence in the accuracy of our MLMF algorithm. The optimum number of HF simulations required to estimate the expected values using MLMC and MLMF is shown in Figure 29 and calculated using (14) and (B7), respectively, with $\epsilon = 3 \times 10^{-2}$. The figure shows that, at all locations, this number decreases as the resolution level increases for both MLMF and MLMC, which is an important result for computational efficiency. An interesting result from Figure 29 is that locations 2 and 8 require the least number of HF simulations in both the MLMC and MLMF algorithms, whilst locations 4 and 5 require the most. When the locations of interest are offshore, and locations 2 and 8 are the furthest offshore, the water elevation there is relatively certain. In contrast, locations 4 and 5 are further inland – at the inlet and behind the dune system – and predicting elevation height at these locations is more uncertain because it is dependent on the amount of overtopping that has occurred, hence larger numbers of simulations are needed to ensure accuracy.

More significantly, Figure 29 shows that MLMF always requires fewer XBeach (HF) simulations than MLMC, with the biggest difference being at level 1, where MLMF requires an order of magnitude fewer simulations. This difference can clearly be seen in Figure 30 which shows the number of HF simulations required by MLMC divided by the number of those required by MLMF. As the level becomes finer, the ratio decreases due to the lower correlation between the two models at the finer levels seen in Figure 26. Nevertheless, even at the finest resolution level, MLMC still requires twice as many simulations as MLMF, which is particularly significant given how computationally expensive the test case is at this resolution (see Table 5). MLMF requires fewer simulations because it uses $r_l N_l^{HF}$ LF simulations. Figure 10 shows that $r_l$ is large at coarse levels but given the computational cost savings from using SFINCS shown in Table 5 this is not an issue. Moreover, as the level number increases and the SFINCS computational cost increases, $r_l$ decreases and is much less than 10 at the finest level.

As in previous test cases, we can apply the modified inverse transform sampling method to the MLMF outputs (here produced using $\epsilon = 3 \times 10^{-2}$) to generate a CDF. As already mentioned, it is impractical to run a Monte Carlo simulation for this test case

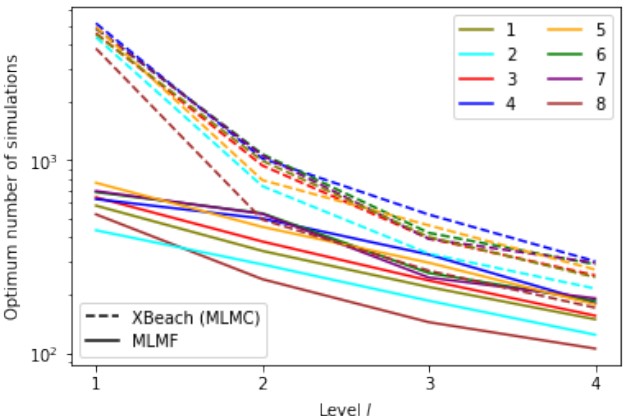

**Figure 29.** Optimum number of XBeach (HF) simulations required by MLMF (Eq. 14) and MLMC (Eq. B7) for the Myrtle Beach test case. Here $\epsilon = 3 \times 10^{-3}$ in (Eq. 14) and (Eq. B7) for MLMF and MLMC, respectively.

**Figure 30.** Optimum number of XBeach (HF) simulations required by MLMC divided by the optimum number required by MLMF for the Myrtle Beach test case. Here $\epsilon = 3 \times 10^{-3}$ in (Eq. 14) and (Eq. B7) for MLMF and MLMC, respectively.

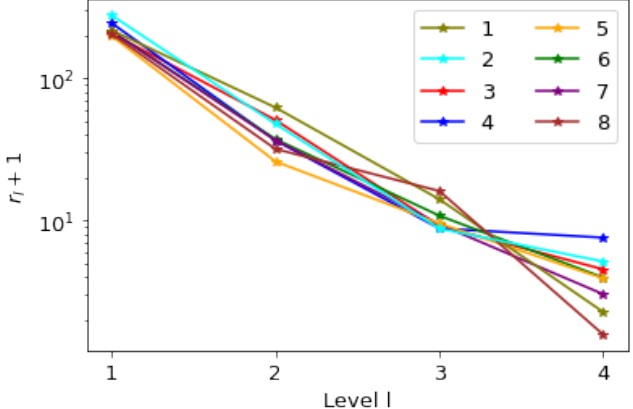

**Figure 31.** Factor of total LF simulations ($r_l + 1$) compared to number of HF simulations, where $r_l$ is (Eq. 13) for the Myrtle Beach test case. Here $\epsilon = 3 \times 10^{-3}$ in (Eq. 14) and (Eq. B7) for MLMF and MLMC, respectively.

and thus, we cannot compare the MLMF-generated CDF with the Monte Carlo-generated CDF as done previously. However the small error norms in Tables 2 and 4 give confidence in the accuracy of the MLMF-generated CDFs for this test case. Figure 32 shows the CDFs for this test case at all eight locations and greatly improves our understanding of the test case. For example, the figure informs that there is a small but significant probability of the maximum elevation height at the inlet (location 5) exceeding 10 m. This is despite the fact that the expected value is only 6.66 m (see Figure 28), which might have led the local

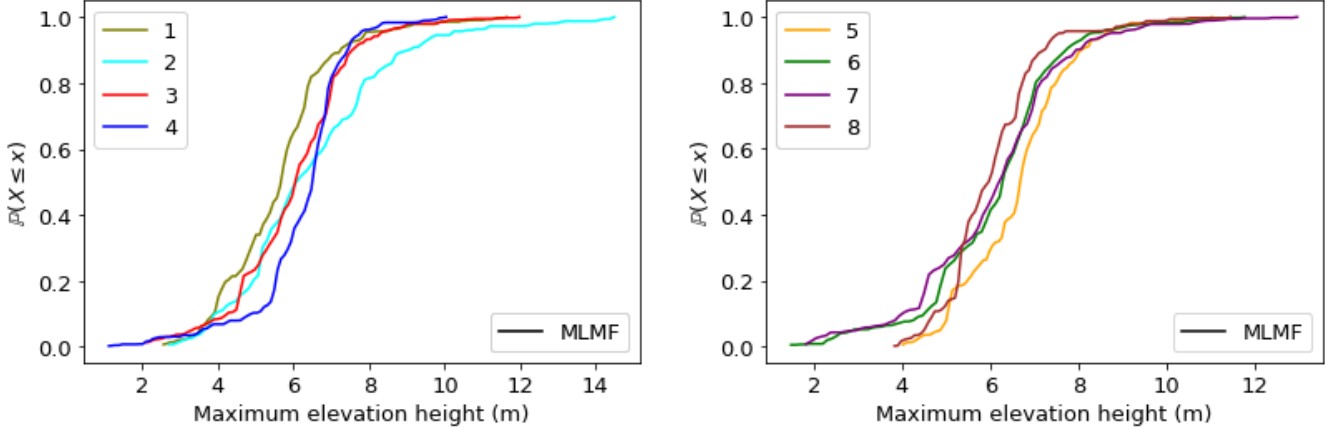

**Figure 32.** CDFs generated from MLMF outputs using the modified inverse transform sampling method (Eq. 17).

authority to believe that they were safe from a 10 m elevation height. This illustrates how important it is for the assessment of the impact of extreme flooding that our MLMF algorithm can accurately and efficiently calculate both the expected values and CDFs of output variables.

Finally, we also consider how different tolerance values $\epsilon$ in (14) affect our expectation results. The trend in the optimum number of HF simulations at each $\epsilon$ (not shown here for brevity) follows that seen in Figure 29: MLMF always requiring fewer simulations than MLMC. Figure 33 shows that the difference in the optimum number of HF simulations required translates to MLMF being more than three times as efficient as MLMC for the same level of accuracy. For such a complex real-world test case, this is a notable result. Although it is impractical to conduct a full analysis using the Monte Carlo method, we have run a Monte Carlo simulation for approximately $3 \times 10^8$ s (1100 simulations). This allows us to conclude that, for the same tolerance, MLMF is over six times faster than Monte Carlo (calculating $\epsilon$ as $\sqrt{2\mathrm{Var}[\cdot]}$ as in the previous test cases). Although this is not a large factor, given the high computational cost, this means that achieving a tolerance of 0.03 takes an estimated 40 years of computational time using the Monte Carlo method compared to only 6 years of computational time using MLMF. Furthermore, unlike MLMC and MLMF, Monte Carlo must always be run for longer than strictly necessary to ensure convergence (see Section 3.1 and 3.2). Therefore, this test case concretely demonstrates that applying MLMF means we can conduct uncertainty analysis of complex real-world problems in an accurate and efficient manner that would have been unfeasible using the standard Monte Carlo method.

## 4 Discussion: Future Extensions to our MLMF methodology

This works aims to be a proof-of-concept demonstrating that MLMF can be used for coastal flooding. Thus, whilst in real-world cases there will be more than one uncertain input, to meet this aim it is sufficient to consider only one uncertain input parameter per test case. Adding more uncertain inputs would increase the variance of the outputs and thus all methods would

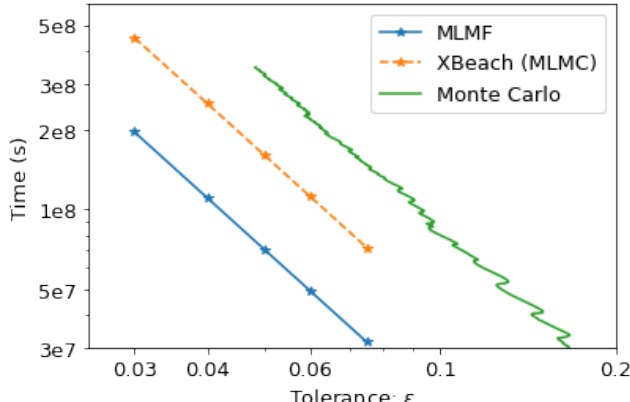

**Figure 33.** Comparing the computational cost required to achieve tolerance $\epsilon$ at all locations using MLMF, XBeach with MLMC and Monte Carlo for the Myrtle Beach test case

require larger numbers of simulations and be more computationally expensive. Note, however, that the methodology outlined in Section 2 remains the same irrespective of the number of uncertain inputs and thus considering multiple uncertain inputs will be the subject of future work.

Furthermore, for all methods in this work, we assess the impact of uncertain input parameters by randomly sampling values from a user-chosen distribution and then running the models with these parameter values. This again meets the aim of this work but is the simplest sampling approach. Nevertheless, the flexibility of MLMC and MLMF means that they can also be combined with other more sophisticated sampling techniques that can further reduce the number of model simulations needed. These complex techniques are out of scope for this work but we remark briefly upon them here. One such technique is Latin hypercube sampling (McKay et al., 2000) which splits the distribution into $n$ equal partitions (where $n$ is the number of samples required) and a sample is then taken from each partition. This sampling approach has been shown to improve computational efficiency when used with both a standard Monte Carlo method (McKay et al., 2000) and with MLMC (Xiong et al., 2022). Another technique is evolutionary algorithms (Vikhar, 2016), which are optimisation algorithms inspired by biological evolution that start with an initial set of samples (population) and evolve towards an optimal set. These have also been successfully combined with MLMC in Pisaroni et al. (2019) to further improve efficiency.

There are also other common techniques to improve the efficiency of assessing uncertainty such as the Markov Chain Monte Carlo method (MCMC) and using machine learning techniques as emulators. As with the sophisticated sampling techniques, these can also be combined with MLMC and/or MLMF to improve the methods further: both multilevel Markov Chain Monte Carlo algorithms (Dodwell et al., 2019) and combining multifidelity samples with transfer learning to train machine learning emulators (Chakraborty, 2021) are fast growing areas of research, making them a promising avenue for further work.

We conclude this section by observing that, although there are more sophisticated techniques to assess uncertainty than that applied in this work, the flexibility of the MLMF algorithm means that it can easily be combined with other more complex

statistical approaches, leveraging the advantages of both approaches. Whilst these combined approaches are beyond the scope of this work, using these techniques on coastal problems is an interesting and promising avenue for further research.

## 5 Conclusions

In this work, we have presented the first successful application of MLMF in the coastal engineering field and one of the first successful applications of this method in any field. Using both idealised and real-world test cases, we have shown that MLMF can significantly improve the computational efficiency of uncertainty quantification analysis in coastal flooding for the same accuracy compared to the standard Monte Carlo method. In particular, we have demonstrated that this enables uncertainty analysis to be conducted in real-world coastal environments that would have been unfeasible with the statistical methods previously applied in this field. Using our new modified inverse transform sampling technique, we are also able to accurately generate the cumulative distribution function (CDF) for the output variables of interest, which is of great value to decision makers. Furthermore, the expected values and CDFs of output variables can be computed at multiple locations simultaneously with no additional computational cost, demonstrating the flexibility of MLMF. In future work, this will enable the construction of large-scale maps showing the expected value and CDF of variables of interest at all locations in the domain, facilitating accurate and timely decision-making. Furthermore, we have highlighted the benefits of using a multifidelity approach and shown that using SFINCS as an LF model and XBeach as a HF model makes MLMF notably more computationally efficient than MLMC for the same or higher accuracy. Multifidelity approaches thus represent a very rewarding avenue for further research and our new model-independent easily applicable MLMF wrapper written as part of this work will greatly facilitate this research.

Finally, this efficient uncertainty quantification can be used in the future for risk estimation. The latter assumes that the same scenario happens repeatedly over a given time period (e.g. rain events over a year) and requires frequency information (e.g. how many times does location X get flooded per time period). Thus, the information gathered by using MLMF to probabilistically quantify the variation/uncertainty in the different scenarios (e.g. rainfall events), can be used in future work for risk estimation.

*Code availability.* The relevant code for the MLMF framework presented in this work is stored at https://github.com/mc4117/MLMF_coastal.

*Author contributions.* MCAC contributed to conceptualization of the project, formal analysis, investigation, methodology, visualization, writing of the original draft and reviewing/editing of the text. TWBL contributed to conceptualization of the project, methodology, visualization, writing of the original draft and reviewing/editing of the text. RTM and FLMD contributed to conceptualization of the project, and reviewing and editing of the text. CJC and MDP contributed to conceptualization and supervision of the project, and reviewing and editing of the text.

*Competing interests.* The authors declare that they have no conflict of interest.

*Acknowledgements.* MCAC's work was funded through the EPSRC CDT in Mathematics for Planet Earth and grant EP/R512540/1. CJC and MDP acknowledge funding from the Engineering and Physical Sciences Research Council (EPSRC) under grants EP/L016613/1, EP/R007470/1 and EP/R029423/1. TWBL and RTM acknowledge funding from the Deltares research programme 'Natural Hazards' and FLMD acknowledges funding from the Deltares research programme 'Risk management'.

**Appendix A: Multifidelity estimators**

Generally, a multifidelity approach uses a low fidelity model to generate surrogate approximations for the outputs of a high fidelity model. If applied correctly, the resulting multifidelity estimator is then as accurate as the equivalent high fidelity one. There exist a number of different multifidelity approaches (see Peherstorfer et al., 2018). MLMF uses the control variate approach which we outline here following Geraci et al. (2015) throughout. The multifidelity estimator is unbiased and given

by

$$\hat{Q}_{M,N}^{HF,CV} = \hat{Q}_{M,N}^{HF} + \alpha_{\text{F}} \left( \hat{Q}_{M,N}^{LF} - \mathbb{E}[Q_M^{LF}] \right), \tag{A1}$$

where $\hat{Q}_{M,N}^{HF,CV}$ is the estimator of the the expected value of a variable of interest, $\mathbb{E}[\cdot]$ denotes expectation, $M$ indicates the fixed discretisation level, and $\alpha_{\text{F}}$ is a scalar. The value of $\alpha_{\text{F}}$ is determined by minimising the variance of $\hat{Q}_{M,N}^{HF,CV}$ and is given by

$$\alpha_{\text{F}} = -\rho \sqrt{\frac{\text{Var}(\hat{Q}_{M,N}^{HF})}{\text{Var}(\hat{Q}_{M,N}^{LF})}}, \tag{A2}$$

where $\rho$ is the Pearson's correlation coefficient for the HF and LF estimators.

Equation (A1) assumes that $\mathbb{E}\left[Q_M^{LF}\right]$ is known, but this is almost never true because we do not know the analytical formula of the distribution of the variable of interest $Q_M^{LF}$. Therefore, extra simulations of the LF model must be conducted in order to estimate this quantity, with its number denoted by $\Delta^{LF}$. Even though we use the same random numbers for the simulations

to construct $\hat{Q}_{M,N}^{HF}$ and $\hat{Q}_{M,N}^{LF}$ (see Figure 2), in the literature the number of simulations is denoted by $N_{HF}$. The number of extra simulations for the LF model is then $\Delta^{LF} = r N_{HF}$, where the optimum value of $r$ is determined later. Thus, the overall computational cost $C$ of the multifidelity estimator is

$$C = C^{HF} + C^{LF}(1+r), \tag{A3}$$

where $C_{HF}$ is the cost of running $N_{HF}$ simulations of the HF model and $C_{LF}$ is the cost of running $N_{HF}$ simulations of the

LF model. Using (A1), the variance 'Var' of the multifidelity estimator is

$$\text{Var}\left[\hat{Q}_{M,N}^{HF,CV}\right] = \text{Var}\left[\hat{Q}_{M,N}^{HF}\right] \left(1 - \frac{r}{1+r}\rho^2\right). \tag{A4}$$

Note that $\rho^2$ is less than one by definition, so $r$ greater than zero means the variance of the estimator is reduced through using this method.

## Appendix B: Multilevel Monte Carlo method (MLMC)

The multilevel Monte Carlo method (MLMC) was first introduced in Giles (2008) and successfully applied in the coastal engineering field in Clare et al. (2021). We refer the reader to those two works for full details of the method and here present a brief overview.

MLMC accelerates the Monte Carlo method by considering the problem at different levels of resolution in a multilevel environment. It then uses linearity of expectations to transform this multi-resolution expectation to a single expectation at the 650 finest level, $L$, using the following formula

$$\mathbb{E}[X_L] = \mathbb{E}[X_{l_\mu}] + \sum_{l=1}^{L} \mathbb{E}[X_l - X_{l-1}]. \tag{B1}$$

Here $X_l$ denotes the numerical approximation to the random variable $X$ on level $l$ of the multilevel environment produced by the model, where in our work $X$ could be the water elevation at a particular location, for example. Thus $X_{l_\mu}$ and $X_L$ denote the approximation on the coarsest ($l_\mu$) and finest level ($L$) respectively. Each level $l$ is defined by its grid-size $h_l$, where

$$h_l \propto M^{-l}T, \tag{B2}$$

and $T$ is the total length of the domain and $M$ the integer factor the grid-size is refined by at each level (following standard practice, we use $M = 2$ throughout). This means that as the level number $l$ increases, the mesh becomes more refined. Trivially, if the domain is multi-dimensional then $T$ and $h_l$ are also multi-dimensional.

Equivalently to (B1), the MLMC expectation estimator $\hat{Y}$ is defined by

$$\hat{Y} = \sum_{l=l_\mu}^{L} \hat{Y}_l, \tag{B3}$$

where $\hat{Y}_l$ is the difference estimator for $\mathbb{E}[X_l - X_{l-1}]$ defined as

$$\hat{Y}_l = \begin{cases} N_{l_\mu}^{-1} \sum_{i=1}^{N_{l_\mu}} X_{l_\mu}^{(i)} & l = l_\mu, \\ N_l^{-1} \sum_{i=1}^{N_l} \left( X_l^{(i)} - X_{l-1}^{(i)} \right) & l > l_\mu. \end{cases} \tag{B4}$$

Here $N_l$ is the number of simulations at each level pair $(l, l-1)$ and $N_{l_\mu}$ is the number of simulations at the coarsest resolution level $l_\mu$. In this estimator, the same random numbers are used to construct the variables $X_l$ and $X_{l-1}$, to ensure strong conver-
gence ($\mathbb{E}[|X_l - X_{l-1}|]$ as the grid is refined). Independence between the estimators at each level is enforced by using different independent samples at each level meaning $\mathrm{Cov}(\hat{Y}_i, \hat{Y}_j) = 0$ if $i \neq j$ and the variance formula can be simplified to

$$\mathrm{Var}[\hat{Y}] = \mathrm{Var}\left( \sum_{l=l_\mu}^{L} \hat{Y}_l \right) = \sum_{l=l_\mu}^{L} N_l^{-1} \mathrm{Var}(\hat{Y}_l), \tag{B5}$$

where Var denotes the variance.

A key factor when using the MLMC estimator is determining the optimum number of simulations to run at each level $l$ denoted by $N_l$. We want to balance the accuracy achieved at the finer levels with the computational efficiency achieved by running at coarser levels. This balance is achieved by following Giles (2008) and using the Euler-Lagrange method to minimize the overall cost $C$ defined by

$$\sum_{l=l_\mu}^{L} N_l C_l, \tag{B6}$$

with respect to the fixed overall variance $\epsilon^2/2$. Thus, the optimum number of simulations at each level is

$$N_l = \left\lceil \frac{2}{\epsilon^2} \sqrt{\frac{\mathrm{Var}(\hat{Y}_l)}{C_l}} \left( \sum_{k=l_\mu}^{L} \sqrt{\mathrm{Var}(\hat{Y}_k)C_k} \right) \right\rceil, \tag{B7}$$

where $C_l$ is the cost of running the model at level $l$ and $\epsilon$ should be seen as a user-defined accuracy tolerance.

However, this formula requires initial estimates of $\mathrm{Var}(\hat{Y}_l)$ and $C_l$ and thus we follow Giles (2008) and run 50 initial simulations (see Step 2 of Algorithm 2). To ensure this provides a good variance estimate, we also calculate the kurtosis (still following Giles (2008)). Following standard practice, if the kurtosis is greater than 100, this indicates that the variance estimate is poor and that the number of initial simulations used is insufficient. In this work, we find 50 is always sufficient but for more complex test cases, a greater number may be required. In our implementation of this algorithm, these initial simulations are stored and used as part of the optimal number of simulations in the final estimator and thus the total cost of running the algorithm is unaffected by these initial simulations (see Step 4 of Algorithm 2).

Note further that when we are estimating multiple outputs (*i.e.* when we consider multiple locations), we must calculate $N_l$ separately for each location. In the algorithm, we run $\max N_l$ over all locations and then when calculating the estimator (B4) at each location, subsample the optimum number for that specific location from the full output.

We have now outlined MLMC and conclude with Algorithm 2, which is a statement of the MLMC algorithm used in this work.

---

**Algorithm 2** Multilevel Monte Carlo Method

---

1: Start with $L = 0$

2: Estimate the variance $\mathrm{Var}(\hat{Y}_L)$ using an initial estimate for the number of simulations $N_L$

3: Define optimal $N_l$ for $l = 0, ..., L$ using (B7)

4: If the optimal $N_l$ is greater than the number of simulations you already have, evaluate the extra simulations needed

5: If $L \geq 2$ test for convergence

6: If $L < 2$ or the algorithm has not converged set $L$ equal to $L + 1$ and return to Step 2

---

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
