# Peer review of "Multilevel multifidelity Monte Carlo methods for assessing uncertainty in coastal flooding"

_Natural Hazards and Earth System Sciences, 2022_

## Author Comment (AC1)

**Multilevel multifidelity Monte Carlo methods for assessing uncertainty in coastal flooding**

*Note from the Authors*
*In this document, we have included the relevant revised sections of the manuscript in framed boxes. Note all references to figures, tables and equations in these boxes, unless otherwise stated, are to the figures and equations in the new manuscript and not to those in this response document.*

*Reviewer 1 comments*

*We would like to thank Reviewer 1 for their helpful and thoughtful comments. Below we outline how we have addressed each of their concerns and suggestions.*

1. The MLMF method introduced by the authors seems to promise a nice way to generate good accuracy within a reasonable runtime for probabasistic hazards. This could be very useful for the nat haz community. However, as a traditionally educated hydrodynamic modeler and coastal engineer, it was very difficult for me to understand the details of the method presented. It would be helpful if the explanation could be extended or revised for understanding by a wider array of possible users. This could include something like "for each SFINCS run we calculate the mean and variance in water levels", "for each XBeach run we calculate mean and variance of water levels", "we do something with each of these means and variances to come up with a total mean and variance". With the explanation given so far, I have difficulty understanding how data from the various levels are used, how the various "samples" are used (there should be a better term for "samples"), and even how the two different models' results are used together. In general, the method should be accessible to hydrodynamic modelers to use, and understandable by such.

   *We will extensively re-write the methodology section in an attempt to make it more accessible to a broad range of coastal engineers and hydrodynamic modellers. As an initial step, we will move the description of the generic methodology for multifidelity estimators and the MLMC method to an appendix and focus solely on the combination MLMF in the main body of the text. We will also improve the introduction to the section with the following prose description of MLMF:*

   > As discussed in Section 1, Monte-Carlo type methods can be used to assess uncertainty by estimating the expectations of functions of an input random variable. In our model scenario, the input random variable is some source of uncertainty, such as the friction coefficient, and the function involves running our numerical model and computing values such as the water elevation height at specific locations, from the model output. These estimates could be calculated using the standard Monte Carlo approach, but this is computationally expensive due to the need to run large numbers of model simulations to obtain an appropriate accuracy (see Eq. 1 and the discussion below it). The computational cost of running the model can be reduced by either coarsening the grid resolution or using a less complex model, or, in the case of this work, making use of both approaches by using the multilevel multifidelity Monte Carlo method (MLMF).
   > Using a coarse grid and/or simpler model gives an estimate which is cheap to compute but (more) incorrect and thus has an error. This error can be corrected by estimating the difference between the low and high fidelity models and/or the different resolutions, and adding these on to the cheaply computed expectation. Key to the approach is the observation that estimating the difference requires fewer simulations than computing the full estimate, because the variance of the correction is (hopefully) smaller than the variance of the outputs. For the different grid resolutions, the correction is done by the telescoping sum of the multilevel Monte Carlo method (MLMC), while for the different fidelity models, the correction is done by control variate formulae. The challenge is composing these approaches so that we can do both, which is what MLMF seeks to do.

   *We will also introduce a hypothetical scenario to better illustrate MLMF in practical terms and use it to extensively describe certain core concepts such as estimators. Following the reviewer's suggestion, we will also change 'samples' to 'simulations'. The proposed new Section 2 is included as an addendum to this document.*

2. Fig 1. I don't understand why the arrow at the left says "Level 1". Also, the term "number of samples" is confusing. Could be easier to understand as "number of runs" or "number of scenarios executed".

*We acknowledge that the arrow with the levels is confusing. It was meant to indicate the mesh getting finer but the figure actually is clearer and simpler without the arrow as you suggest so we will remove it. Also following your suggestion we will change number of samples to number of simulations. We have included below the modified figure:*

[Figure]

High fidelity model (XBeach)   Low fidelity model (SFINCS)

$N^o$ *of simulations executed* $- O(10^3)$   $N^o$ *of simulations executed* $- O(10^5)$

€   €
€ € €

$N^o$ *of simulations executed* $- O(10^2)$   $N^o$ *of simulations executed* $- O(10^4)$

€ €   € €
€ € €

$N^o$ *of simulations executed* $- O(10^1)$   $N^o$ *of simulations executed* $- O(10^3)$

€ € €   € € €
€ € €

Figure 1: *Example illustration of how MLMF's multifidelity multilevel approach using SFINCS and XBeach models on different grid resolutions results in computational cost savings. Note the € symbol indicates the order of magnitude of the computational cost for a single simulation with this model at this grid resolution i.e. €€ indicates $O(10^2)$ seconds for a single simulation. The orders of time and number of scenarios are approximately those for the Myrtle Beach test case in Section 3.3.*

3. Line 163. What does := mean?

*:= means equal by definition, or is defined to be, but we will make this line simpler by changing it to the following*

set $L$ equal to $L + 1$

4. Please explain clearly the difference between Eq. 3 and Eq. 4.

*The first equation is the generic formula and the second equation is the actual expression of the estimator. We acknowledge this is confusing and therefore will completely remove the first equation (Eq 2.) and keep only the second equation (Eq 3.). Note, we have answered this query assuming you are referring to Eq 2 and 3, rather than 3 and 4.*

5. It is difficult to tell whether sections 2.1, 2.2, and 2.3 are all just re-explaining what Geraci et al. 2015 showed, or whether this is new material. Please clarify this in the text.

*Section 2.1 is indeed largely re-explaining what Geraci et al. 2015 showed; we will add the following text at the beginning of this sub-section:*

In this section, we describe the theory for the standard MLMF approach, following Geraci et al. (2015) throughout.

*Section 2.2 is a novel technique which we have developed as a part of this work and therefore we will add the following sentence at the beginning of this section:*

> To resolve this, in this work we develop our own novel technique to find the cumulative distribution function (CDF) from the MLMF outputs, using a modified version of the inverse transform sampling method from Gregory and Cotter (2017).

*Section 2.3 is also new material and therefore we will add the following sentence at the beginning of this section:*

> In this work, we construct our own Python MLMF wrapper around both SFINCS and XBeach to implement the MLMF algorithm.

6. Why doesn't RMSE have a unit? Does RMSE mean relative RMSE, normalized to be unitless? This should be stated clearly.

*We will add the units for the RMSE to the figures (see below for example which is a recreation of the updated figure 7). We will also add the following line to the error norm tables (2 and 4) to emphasise that these errors are unitless.*

[revised manuscript text omitted]

---

## Author Comment (AC2)

**Multilevel multifidelity Monte Carlo methods for assessing uncertainty in coastal flooding**

***Note from the Authors***

*In this document, we have included the relevant revised sections of the manuscript in framed boxes. Note all references to figures, tables and equations in these boxes, unless otherwise stated, are to the figures and equations in the new manuscript and not to those in this response document.*

***Reviewer 2 comments***

*We would like to thank Reviewer 2 for their helpful and thoughtful comments. Below we outline how we have addressed each of their concerns and suggestions.*

1. In all, the methodology, as presented, seems to be more adequate for conducting uncertainty quantification analysis and not for coastal flood risk assessment. Not much evidence is presented in the paper supporting the usefulness of the methodology in risk assessment. The estimation of cumulative distribution functions (CDFs) is just one piece of a proper risk assessment. In areas like Myrtle Beach, SC, and throughout most of the U.S. Atlantic and Gulf coasts, risk assessment must be concerned with understanding and characterizing coastal hazards (e.g., storm surge, waves, tides) due to different storm populations (e.g., tropical cyclones, extratropical storms), and must rely on some form of extreme value analysis. None of these elements are present in this paper. A methodology developed for uncertainty analysis is not necessarily transferable to coastal hazard analysis or risk assessment, therefore, I would consider revising the title of the paper as it might be inadvertently misleading.

   *As suggested, we will remove all references to risk assessment in the manuscript and replace them with references to uncertainty. This includes the title for which we propose the following:*

   > Multilevel multifidelity Monte Carlo methods for assessing uncertainty in coastal flooding

2. Although my background and experience encompass both probabilistic hazard analysis and hydrodynamic modeling, I found this paper to be quite difficult to follow. The abstract states: "Here, we apply the multilevel multi-fidelity Monte Carlo method (MLMF) to quantify uncertainty by computing statistical estimators of key output variables with respect to uncertain inputs, ..."; but there is no discussion about how are these uncertain inputs identified or prioritized. The three cases presented in the manuscript, including 2D real-world case, are highly idealized and seem to consider only one uncertain input per case; this is, Manning's coefficient, beach slope, and offshore water level, respectively.

   *We will rewrite Section 2 to make it simpler to follow the methodology. The proposed new Section 2 is included as an addendum to this document.*

   *We will also add text to each test case explaining why each uncertain parameter was selected. For the first case, we will add the following:*

   > We choose the Manning coefficient as our uncertain parameter because Bates et al. (2010) note that this test case is particularly sensitive to this parameter and thus this is a good test for our MLMF framework.

   *For the second test case, we will add the following:*

   > We choose the slope as our uncertain parameter because it represents a significant source of uncertainty, as discussed in Unguendoli (2018), particularly when simulating run-up and run-down as is the case here.

   *For the third test case we will add the following:*

   > Over the coming decades, climate change will lead to changing water levels but the actual change at specific locations is uncertain, which in turn leads to uncertainty in the impact of flooding from future storms. Thus, in this test case, we consider the offshore water level to be uncertain.

3. In real-world applications, rarely there is just one uncertain input parameter. The paper discusses how to consider multiple output locations, but is the methodology applicable when there are multiple uncertain input parameters?

*The methodology is the same irrespective of the number of uncertain input parameters. Whilst we agree with the comment that there is normally more than one uncertain parameter, we have chosen to only use one uncertain input throughout, largely because increasing the number of uncertain inputs will increase the general uncertainty of the test case meaning larger numbers of samples are required by the Monte Carlo, the MLMC and the MLMF methods. We will add a new discussion section before the conclusion, where we discuss this, amongst other things (see response to comment 5).*

**Discussion: Future Extensions to our MLMF methodology**

This works aims to be a proof-of-concept demonstrating that MLMF can be used for coastal flooding. Thus, whilst in real-world cases there will be more than one uncertain input, to meet this aim it is sufficient to consider only one uncertain input parameter per test case. Adding more uncertain inputs would increase the variance of the outputs and thus all methods would require larger numbers of simulations and be more computationally expensive. Note, however, that the methodology outlined in Section 3 remains the same irrespective of the number of uncertain inputs and thus considering multiple uncertain inputs will be the subject of future work.

4. The initial case (presumably Level 0) does not seem to be well defined. Equation #14 is used to determine the optimal number of samples, but how can the initial number of samples be estimated without prior knowledge of key output variance and the input-to-output relationship?

*It is indeed necessary to run an initial number of samples to estimate variance and cost. This is already indicated in Step 2 of the MLMC algorithm (Algorithm 1) and Step 1 of the MLMF algorithm (Algorithm 2), but we will add text to highlight this. After equation 14, we will also add the following:*

However, this formula requires initial estimates of $\text{Var}(\hat{Y}_l)$ and $C_l$ and thus we follow Giles (2008) and run 50 initial simulations (see Step 2 of Algorithm 1). To ensure this provides a good variance estimate, we also calculate the kurtosis (still following Giles (2008)). Following standard practice, if the kurtosis is greater than 100, this indicates that the variance estimate is poor and that the number of initial simulations used is insufficient. In this work, we find 50 is always sufficient but for more complex test cases, a greater number may be required. In our implementation of this algorithm, these initial simulations are stored and used as part of the optimal number of simulations in the final estimator and thus the total cost of running the algorithm is unaffected by these initial simulations (see Step 4 of Algorithm 1).

5. Also, there is no mention of other approaches that are used for similar purposes, this is, to determine the optimal number of events to be subsequently simulated using high fidelity models. Such approaches could leverage methods like Latin hypercube sampling, genetic algorithms, joint probability methods, and even recent machine learning techniques that directly account for input-output relationships. It's difficult to judge the benefits of the proposed methodology without a discussion, at least conceptually, of some of these other approaches.

*We will add a new discussion section before the conclusion (see response to comment 3), where we will acknowledge other alternative approaches. We note here and in the text that many of these approaches can be combined with MLMF to improve upon existing approaches.*

For all methods in this work, we assess the impact of uncertain input parameters by randomly sampling values from a user-chosen distribution and then running the models with these parameter values. This again meets the aim of this work but is the simplest sampling approach. Nevertheless, the flexibility of MLMC and MLMF means that they can also be combined with other more sophisticated sampling techniques that can further reduce the number of model simulations needed. These complex techniques are out of scope for this work but we remark briefly upon them here. One such technique is Latin hypercube sampling (McKay et al., 2000) which splits the distribution into $n$ equal partitions (where $n$ is the number of samples required) and a sample is then taken from each partition. This sampling approach has been shown to improve computational efficiency when used with both a standard Monte Carlo method (McKay et al., 2000) and with MLMC (Xiong et al., 2022). Another technique is evolutionary algorithms (Vikhar, 2016), which are optimisation algorithms inspired by biological evolution that start with an initial set of samples (population) and evolve towards an optimal set. These have also been successfully combined with MLMC in Pisaroni et al. (2019) to further improve efficiency. There are also other common techniques to improve the efficiency of assessing uncertainty such as the Markov Chain Monte Carlo method (MCMC) and using machine learning techniques as emulators. As with the sophisticated sampling techniques, these can also be combined with MLMC and/or MLMF to improve the methods further: both multilevel Markov Chain Monte Carlo algorithms (Dodwell et al., 2019) and combining multifidelity samples with transfer learning to train machine learning emulators (Chakraborty, 2021) are fast growing areas of research, making them a promising avenue for further work.

We conclude this section by observing that, although there are more sophisticated techniques to assess uncertainty than that applied in this work, the flexibility of the MLMF algorithm means that it can easily be combined with other more complex statistical approaches, leveraging the advantages of both approaches. Whilst these combined approaches are beyond the scope of this work, using these techniques on coastal problems is an interesting and promising avenue for further research.

6. Several numerical models are introduced in the abstract and the manuscript introduction without defining the name (or acronym); e.g., SFINCS is not defined (Super-Fast INundation of CoastS) until the third time that the model is mentioned.

*We will add a definition of SFINCS to the abstract and move the definition in the Introduction to where the first time SFINCS is mentioned. We will also define the acronyms for the other models where they first appear. The sentence where different high and low fidelity models are introduced in the introduction will read as follows:*

Coastal flood modelling is therefore an ideal field on which to apply MLMF because there exist a large number of high fidelity but computationally expensive full physics models such as XBeach (Roelvink et al., 2009), SWASH (Simulating WAves till SHore) (Zijlema et al., 2011), or MIKE21 (Warren and Bach, 1992), and lower fidelity computationally cheaper reduced physics models such as SFINCS (Super-Fast INundation of CoastS) (Leijnse et al., 2021), LISFLOOD-FP (Bates et al., 2010) or SBEACH (Storm-Induced BEAch CHange) (Larson and Kraus, 1989).

7. Figure 1 mentions "Level 1", but the concept of what a level constitutes in this methodology has not been established at this stage of the manuscript. Also, in Figure 1, is the Euro symbol meant to describe typical computational costs or time associated with the different levels of fidelity and number of samples? It might help to provide additional context. In terms of order of magnitude, is each "Euro symbol" representing hours, weeks, or months?

*We had not noticed that the concept of a level had not yet been established by Figure 1 and thus following your helpful suggestion, we will remove all references to levels in this figure. We will also improve the caption to explain the "Euro symbol" better.*

[Figure]

*Figure 1: Example illustration of how MLMF's multifidelity multilevel approach using SFINCS and XBeach models on different grid resolutions results in computational cost savings. Note the € symbol indicates the order of magnitude of the computational cost for a single simulation with this model at this grid resolution i.e. €€ indicates $O(10^2)$ seconds for a single simulation. The orders of time and number of scenarios are approximately those for the Myrtle Beach test case in Section 3.3.*

[revised manuscript text omitted]

---

## Author Response (AR2)

I confirm Figure 24 was entirely created by the authors and there is no need to add a copyright or credit statement.